# Arylmethylamino steroids as antiparasitic agents

Reimar Krieg[1], Esther Jortzik[2], Alice-Anne Goetz[3], Stéphanie Blandin[3], Sergio Wittlin[4,5], Mourad Elhabiri[6], Mahsa Rahbari[2], Selbi Nuryyeva[6,7], Kerstin Voigt[8], Hans-Martin Dahse[8], Axel Brakhage[8], Svenja Beckmann[9], Thomas Quack[9], Christoph G. Grevelding[9], Anthony B. Pinkerton[10,11], Bruno Schönecker[12], Jeremy Burrows[13], Elisabeth Davioud-Charvet[6], Stefan Rahlfs[2] & Katja Becker[2]

In search of antiparasitic agents, we here identify arylmethylamino steroids as potent compounds and characterize more than 60 derivatives. The lead compound **1o** is fast acting and highly active against intraerythrocytic stages of chloroquine-sensitive and resistant *Plasmodium falciparum* parasites (IC$_{50}$ 1–5 nM) as well as against gametocytes. In *P. berghei*-infected mice, oral administration of **1o** drastically reduces parasitaemia and cures the animals. Furthermore, **1o** efficiently blocks parasite transmission from mice to mosquitoes. The steroid compounds show low cytotoxicity in mammalian cells and do not induce acute toxicity symptoms in mice. Moreover, **1o** has a remarkable activity against the blood-feeding trematode parasite *Schistosoma mansoni*. The steroid and the hydroxyarylmethylamino moieties are essential for antimalarial activity supporting a chelate-based quinone methide mechanism involving metal or haem bioactivation. This study identifies chemical scaffolds that are rapidly internalized into blood-feeding parasites.

[1] Institute of Anatomy II, University Hospital Jena, Teichgraben 7, 07743 Jena, Germany. [2] Biochemistry and Molecular Biology, Interdisciplinary Research Centre, Justus Liebig University Giessen, Heinrich Buff Ring 26-32, 35392 Giessen, Germany. [3] Université de Strasbourg, CNRS, Inserm, MIR UPR9022/U963, F-67000 Strasbourg, France. [4] Swiss Tropical and Public Health Institute, Socinstrasse 57, PO Box, 4002 Basel, Switzerland. [5] University of Basel, Petersplatz 1, 4001 Basel, Switzerland. [6] UMR 7509 Centre National de la Recherche Scientifique and University of Strasbourg, European School of Chemistry, Polymers and Materials (ECPM), 25, rue Becquerel, F-67087 Strasbourg, France. [7] New York University Abu Dhabi, PO Box 129188, Abu Dhabi, UAE. [8] Leibniz Institute for Natural Product Research and Infection Biology-Hans Knöll Institute (HKI), Adolf-Reichwein-Strasse 23, 07745 Jena, Germany. [9] Institute of Parasitology, Biomedical Research Centre, Justus Liebig University Giessen, Schubertstrasse 81, 35392 Giessen, Germany. [10] Conrad Prebys Center for Chemical Genomics, Sanford-Burnham-Prebys Medical Discovery Institute, 10901 North Torrey Pines Road, La Jolla, California 92037, USA. [11] Conrad Prebys Center for Chemical Genomics, Sanford-Burnham-Prebys Medical Discovery Institute, 6400 Sanger Road, Orlando, Florida 32827, USA. [12] Institute of Organic and Macromolecular Chemistry, Friedrich Schiller University Jena, Humboldtstrasse 10, 07743 Jena, Germany. [13] Medicines for Malaria Venture, 20 Route de Pré-Bois, 1215 Geneva 15, Switzerland. Correspondence and requests for materials should be addressed to K.B. (email: katja.becker@uni-giessen.de).

Malaria caused by the unicellular apicomplexan parasite *Plasmodium* still threatens about 3.2 billion people worldwide. In 2015 there were an estimated 214 million cases of malaria and 438,000 deaths[1]. Artemisinin-based combination therapies are the currently recommended treatment for uncomplicated *P. falciparum* malaria[1], however, parasites exhibiting artemisinin-resistance have now been reported[2,3]. Thus, the discovery and development of novel antimalarials and transmission-blocking agents is of global importance[4,5]. Schistosomiasis (bilharzia) ranks second to malaria as a parasitosis affecting more than 240 million people in the tropics and subtropics[6]. A vaccine against these parasites is not yet available, and praziquantel (PZQ) is the only commonly used drug for treatment, justifying the fear of upcoming resistance[7].

In connection with our work on steroids as chiral ligands for metal ions[8] we became interested in the biological activity of steroid compounds. In general, steroids possess a conformation of their lipophilic framework, which allows the placement of substituents in well-defined spatial environments and conformational freedom. Also, there is a great structural variability of the tetracyclic framework, including estratrienes, androstanes, and cholestanes. In addition, ring junctions (5α/5β, 14α/14β and so on) can differ considerably. Functional groups can be attached at up to 27 positions in the framework or on side chains, and advances in synthetic chemistry over the last several years have made accessing many of these derivatives possible. In addition to their use in pharmaceuticals, there has also been growing interest in the use of steroid derivatives as rigid homochiral model compounds, for example, biomimetics, or as templates for chemo-, regio- and stereoselective investigation. Thus, one approach to discover new biologically active compounds is to combine a steroid skeleton with structural elements possessing appropriate biological activities.

Interestingly, during a screening program for potential antiprotozoal drugs, the naturally occurring steroid 3β-amino-22,26-epiminocholest-5-ene (sarachine) from the leaves of *Saraca punctate,* had been isolated as a hit. Despite its simple substitution patterns, it was reported to exhibit activity against the malaria parasite *Plasmodium falciparum*[9], and a series of amino steroids having side chains similar to that of sarachine were prepared from deoxycholic acid as the starting material[10]. To this end, the most active derivative of this series contained a chloroquinoline moiety in the side chain, which might be contributing to biological activity. The advantage of employing hydrophobic steroid units is their membrane permeability, paving the way for biologically active hybrid molecules. On the basis of this knowledge ω-pyridiniumalkyl ethers of steroidal phenols were synthesized and indeed the combination of the hydrophobic steroid moiety with a hydrophilic group (ferrocenylmethylamino group, *N*-alkyl-pyridinium groups)[11,12] led to compounds with antimicrobial activity. In continuation of this work, the 3-methoxy-estra-1,3,5(10)-triene unit was combined with the *o*-hydroxybenzylamino group[13] and resulted in compounds with antimalarial activity which were systematically optimized using structure-activity relationship approaches.

Here we report on arylmethylamino steroids with excellent antimalarial activity. The compounds are active *in vitro* against human and murine *Plasmodium* asexual blood stages as well as against *P. falciparum* gametocytes. Furthermore they exhibit *in vivo* activity in *Plasmodium berghei*-infected mice and block the transmission of parasites to mosquitoes. Notably the compounds also have fatal effects on adult *Schistosoma mansoni*. The lipophilic steroid carrier of these antiparasitic lead compounds is likely to facilitate membrane permeation and bioavailability whereas the essential hydroxyarylmethylamino moiety points to a chelate-based quinone methide mechanism involving metal or haem bioactivation.

## Results

**Chemistry**. The low molecular weight steroid compounds described here are based on a substituted steroidal pharmacophore and represent novel chemical matter[14]. All compounds are derived from amino steroids with varying constitutions of the basic gonane core (series **1–14**) and varying functional groups R (substitution patterns **c**, **e**, **g**, **i**, **k**, **m**, **q**, **s**, **u** and **w**, Fig. 1). In brief, the compounds were synthesized via condensation of amino steroids with arylaldehydes, and subsequent $NaBH_4$-reduction of obtained *Schiff*-bases furnished the desired compounds in high yields. The synthesis could also be carried out as a one-pot procedure without isolation of *Schiff*-bases. This is especially beneficial in case of *cis*-configured amino alcohols (series **5**, **8**, **11** and **14**) where reaction with aldehydes leads to rapidly exchanging tautomeric *Schiff*-based/oxazoline equilibria. Complete synthetic details can be found in Supplementary Note 1, [1]H and [13]C NMR spectra of the most relevant compounds are provided in Supplementary Figs 1–6.

**Activity against *Plasmodium falciparum* blood stages *in vitro***. The activity of the steroid compounds was tested against the asexual blood stages of the malaria parasite *P. falciparum* (strain 3D7) in the [3]H-hypoxanthine incorporation assay. As shown in Table 1, the compounds are highly active against red blood cell stages of *P. falciparum* with $IC_{50}$ values in the nanomolar range. The most active compound **1o** had an $IC_{50}$ value of $4.1 \pm 1.6$ nM on 3D7 ($n = 6$), ($\pm$ indicates s.d. throughout the manuscript, unless otherwise stated) which is comparable to the *in vitro* activity of clinically employed antimalarials such as chloroquine ($IC_{50} = 8.6$ nM) and artemisinin ($IC_{50} = 17.3$ nM) using the same methods (72 h [3]H-hypoxanthine incorporation assay)[15]. The second most active compound **2o** showed an $IC_{50}$ value of $6.6 \pm 2.1$ nM ($n = 4$) on 3D7 (for data on all compounds tested, please see Supplementary Table 1). Notably, the compounds were found to be more active against chloroquine-resistant parasites as tested for the *P. falciparum* strain Dd2: **1o**: $1.0 \pm 0.9$ nM ($n = 4$); **2o**: $2.0 \pm 1.2$ nM ($n = 4$); chloroquine: 90.2 nM; artemisinin: 20.4 nM (ref. 15) (Supplementary Fig. 7). A similar phenomenon has been reported for other antimalarial compounds as diverse as 8-aminoquinolines and amadantine. The underlying mechanism of this very promising activity of the steroid compounds deserves further attention.

To assess the rapidity of growth inhibition induced by the steroid compounds, the so-called $IC_{50}$ speed assay[16] was conducted for compound **1o** on *P. falciparum* NF54 (Fig. 2a,b). In this assay, a comparison of $IC_{50}$ values, which are obtained side-by-side after 72, 48, and 24 h of compound incubation, is carried out (the 72 h assay is the standard $IC_{50}$ assay otherwise used in our study). For compound **1o**, no significant $IC_{50}$ shifts were determined across all time points, indicating that the compound is fast acting like chloroquine or artesunate and not slow acting like pyrimethamine.

To further study the mechanism of action of the steroid compounds and evaluate the potential involvement of redox stress, compound **1o** was tested in cell culture on the chloroquine-sensitive *P. falciparum* 3D7 strain expressing the cytosolic glutathione redox sensor hGrx1-roGFP2 (Supplementary Note 2). As indicated in Supplementary Fig. 8, 24 h incubation with 20, 50 and 100 nM of compound **1o** led to a dose-dependent increase of the redox ratio of the probe, pointing to increased oxidation and alterations in the intracellular

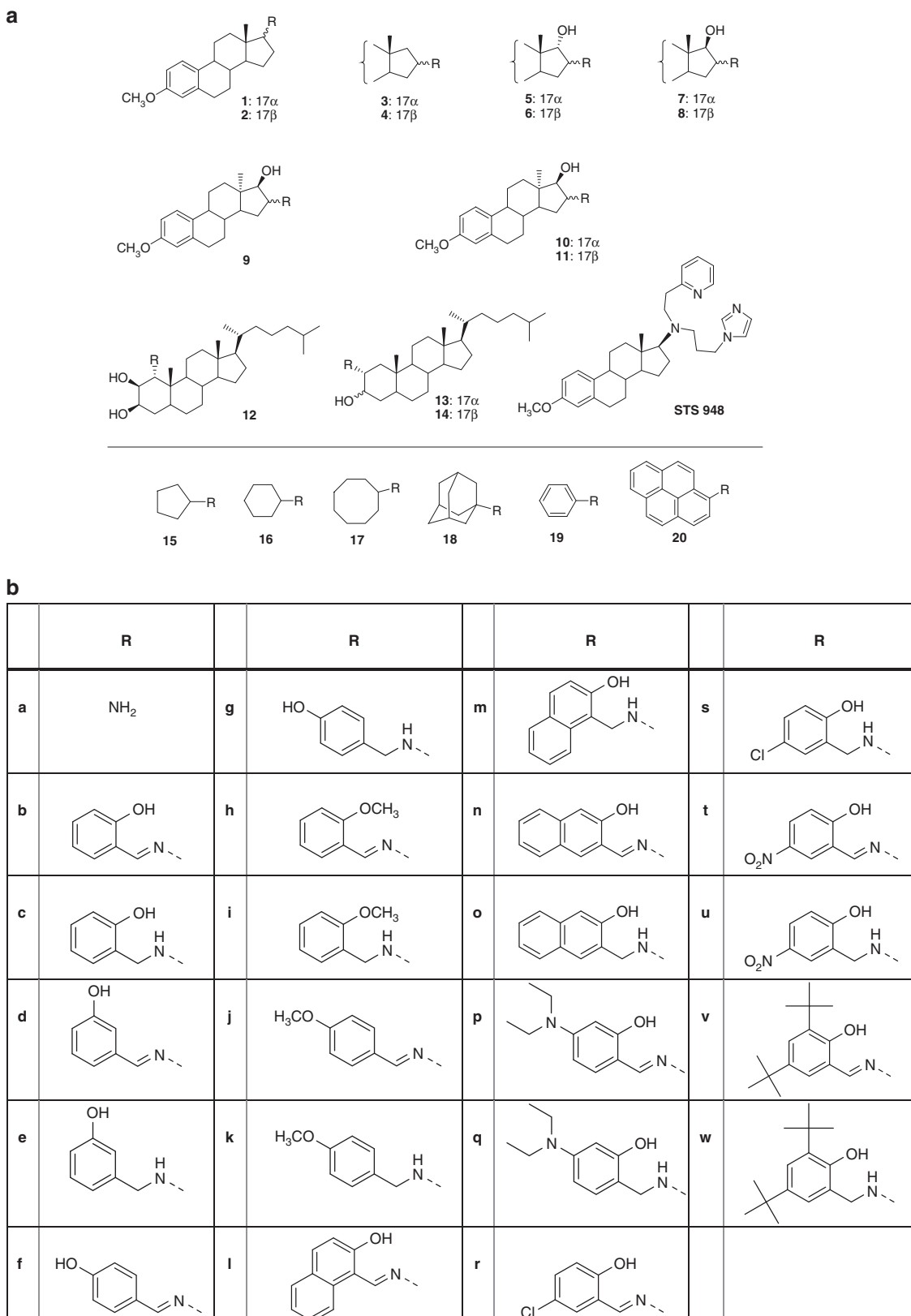

**Figure 1 | Formulas of compounds investigated.** (**a**) Basic structures and (**b**) corresponding substituents.

redox potential. Although these results fell short in reaching significance in the experimental setup chosen, they point towards a potential influence of compound **1o** on the glutathione redox homeostasis, which deserves to be studied in more detail.

**Activity against *P. berghei in vivo* and acute toxicity.** Compounds with the highest activity in the *P. falciparum* cell culture were tested in an *in vivo* mouse model (*P. berghei*) at the Swiss TPH as previously described[17–19] (Supplementary Table 2).

**Table 1 | Antiplasmodial effects as well as antiproliferative and cytotoxic activity of selected steroid compounds on mammalian cells.**

| Compound | Antiplasmodial activity | Antiproliferative activity | | Cytotoxicity |
| --- | --- | --- | --- | --- |
| | $IC_{50}$ [nM] | $GI_{50}$ [µM] | | $CC_{50}$ [µM] |
| | Pf (3D7) | HUVEC $GI_{50}$ | K-562 $GI_{50}$ | HeLa $CC_{50}$ |
| **1c** | 7.4 | 17.1 | 5.9 | >128 |
| **1o** | 4.1 | 108.5 | 5.0 | >113 |
| **2c** | 73 | 10.0 | 5.1 | 42.9 |
| **2o** | 6.6 | 13.8 | 2.5 | >113 |
| **3c** | 42 | 4.1 | 3.8 | 20.7 |
| **4c** | 63 | 3.1 | 2.6 | 12.0 |
| **5c** | 160 | 2.2 | 2.0 | 11.3 |
| **6c** | 676 | 4.4 | 2.5 | 11.3 |
| **7c** | 120 | 1.2 | 2.0 | 10.1 |
| **8c** | 224 | 2.5 | 0.5 | 5.9 |
| **9c** | 999 | 11.2 | 8.4 | 48.8 |
| **10c** | 84 | 2.9 | 2.5 | 11.5 |
| **11c** | 141 | 22.0 | 0.1 | 6.4 |
| **12c** | 4,360 | 12.4 | 16.5 | 38.0 |
| **13c** | 465 | 23.7 | 17.7 | 85.3 |
| **14c** | 208 | 6.7 | 8.2 | 21.8 |
| **STS 948** | 551 | 7.2 | 6.6 | 18.3 |

Strong antiproliferative effects are judged to occur at a $GI_{50}$: ≤2 µM. All values are mean values of at least three independent determinations that differed by less than 20%. SDs for antiplasmodial activity are provided.

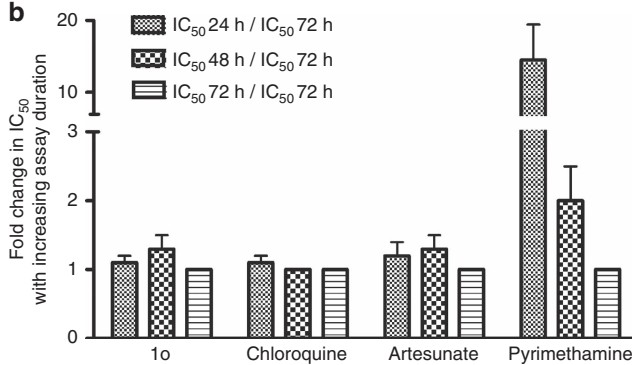

**a**

| | $IC_{50}$ 24 h (nM) | $IC_{50}$ 48 h (nM) | $IC_{50}$ 72 h (nM) |
| --- | --- | --- | --- |
| **1o** | 2.9 ± 0.2 | 3.4 ± 0.2 | 2.7 ± 0.4 |
| Chloroquine | 8.3 ± 2.1 | 7.5 ± 0.9 | 7.4 ± 1.1 |
| Artesunate | 5.5 ± 0.8 | 5.7 ± 0.5 | 4.7 ± 0.3 |
| Pyrimethamine | 220 ± 61 | 30.4 ± 7.6 | 15.2 ± 1.6 |

**b**

**Figure 2 | $IC_{50}$ values of 1o and established antimalarial drugs as determined in the $IC_{50}$ speed assay on *P. falciparum* NF54 *in vitro*[18].** (**a**) Provided are mean values ± s.d. from three biological replicates. (**b**) The fold change in $IC_{50}$ over time is indicated, showing that only pyrimethamine is a slow-acting compound.

Initially, the compound **2c** was tested, which exhibited an $IC_{50}$ of 73 nM against *P. falciparum* 3D7 *in vitro*. The compound was active when applied to the mice subcutaneously (s.c.) or orally over three consecutive days. Subcutaneous application of $3 \times 10$ mg kg$^{-1}$ reduced parasitaemia by 47% and increased the life span of the animals from 6 days to 10 days (parasitized control mice were killed on day 4 to prevent death otherwise occurring at day 6); $3 \times 30$ mg kg$^{-1}$ s.c. reduced parasitaemia by 87% and increased the life span from 6 to

14 days. After oral application of $3 \times 100$ mg kg$^{-1}$ **2c**, parasitaemia decreased by 99%, and survival time increased to 16 days. As shown in Supplementary Table 2 compounds **1c** and **1o** (optimized on the basis of **2c**) were studied in more detail. Compound **1c**, which had an *in vitro* activity against *P. falciparum* 3D7 of 7.4 nM, was also active after s.c. and oral application and showed a similar *in vivo* activity pattern as **2c**. Parasitaemia was drastically decreased, and the life span of the animals was enhanced.

Before carrying out further *P. berghei in vivo* experiments, compound **1o**, which had the best *in vitro* activity against *P. falciparum* 3D7 (72 h incubation, $IC_{50} = 4.1$ nM) and a strong activity against *P. berghei ex vivo* (24 h incubation, $IC_{50} = 3.1 \pm 2.4$ nM ($n = 3$) and methylene blue which served as a control resulted in $2.6 \pm 1.2$ nM ($n = 3$)), were tested in an 'acute toxicity' *in vivo* model. The compound was applied at an accumulative dose of 100 mg kg$^{-1}$ (*per os* (p.o.) as well as intraperitoneally (i.p.)). The first application was 5 mg kg$^{-1}$, 2 h later 15 mg kg$^{-1}$, 2 h later 30 mg kg$^{-1}$ and again 2 h later 50 mg kg$^{-1}$. No acute toxicity symptoms were observed in that experiment. Moreover, we performed pharmacokinetics studies in mice, dosing i.v., i.p. and p.o. (Supplementary Table 5). **1o** displayed moderate clearance (29.6 ml min$^{-1}$ kg$^{-1}$) with a long half-life (>8 h) after i.v. dosing. The compound showed modest oral bioavailability (%F < 5) but sustained plasma levels of ∼100 nM after oral dosing at 100 mg kg$^{-1}$. In contrast, high exposure and sustained levels of ∼2 µM were obtained after i.p. dosing at 100 mg kg$^{-1}$. On the basis of these data, we decided to perform the following *P. berghei in vivo* experiments at doses of $4 \times 100$ mg kg$^{-1}$. After i.p. administration of $4 \times 100$ mg kg$^{-1}$ **1o**, all mice treated were cured; after oral administration two thirds of the animals were cured, which is consistent with the improved exposure after i.p. dosing. Notably, a single dose ($1 \times 100$ mg kg$^{-1}$) p.o. also reduced parasitaemia by 98.46% and increased the life span from 6 to 14 days.

**Effects on transmission and gametocytes.** We further tested the activity of compound **1o** on parasite transmission from mice to *Anopheles gambiae* mosquitoes. For this, *P. berghei*-infected mice

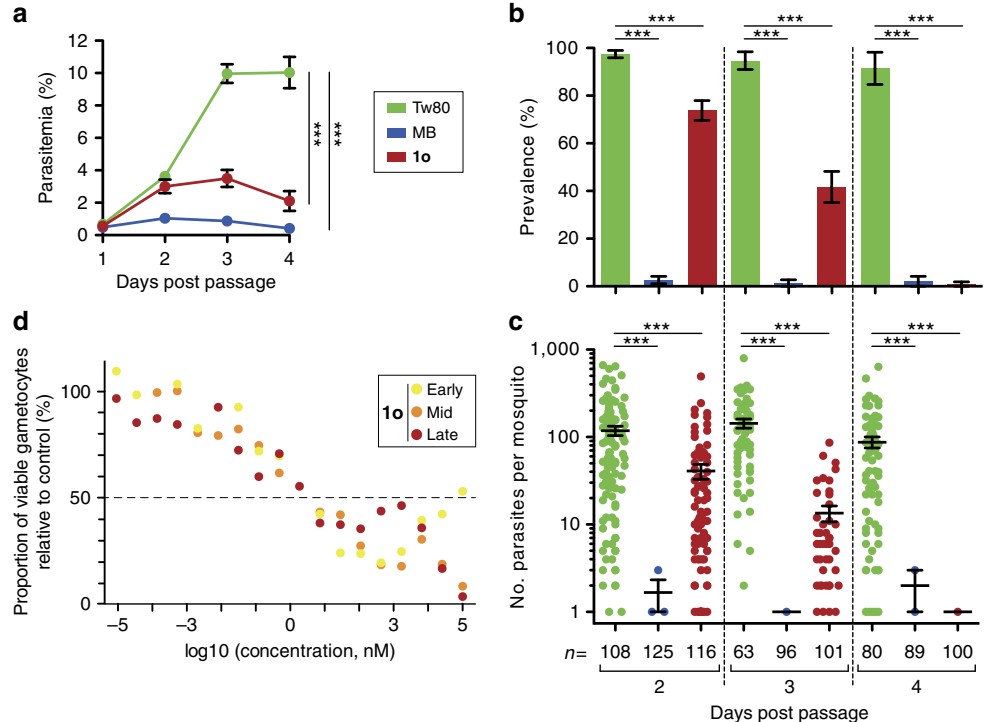

**Figure 3 | Effect of compound 1o on *P. berghei* multiplication in mice and on parasite transmission to mosquitoes.** (**a**) Naive mice were injected i.v. with infected blood and treated daily for 3 days starting 24 h post passage with compound **1o** (100 mg kg$^{-1}$), with methylene blue (MB, 15 mg/kg), or with vehicle (Tw80) as positive and negative controls, respectively. Parasitemia was monitored daily and the mean ± s.d. of three independent experiments are plotted. Groups of mosquitoes were fed on mice 24 h after each treatment. (**b**) Prevalence (percentage of infected mosquitoes) and (**c**) infection intensity (mean number of parasites per infected midgut) were determined 7 days post infectious blood feeding. The results of four independent experiments were pooled, and the mean ± s.e. of the mean is shown (**b,c**). Significance for differences between treatments was calculated using generalized linear models with treatment, time, and repeats as variables, ***$P$ value < 0.001 (**a–c**). $n$, number of mosquitoes. (**d**) Effects of compound **1o** on early, mid, and late stage *P. falciparum* gametocytes. Representative data sets are given (4.5% gametocytaemia) out of three independent experiments. Survival of treated gametocytes was calculated for each concentration relative to parasites exposed to vehicle.

received a daily injection of compound **1o** for 3 days starting 24 h post passage and were exposed to mosquito bites 24 h after each injection. Tween 80 (Tw80) and methylene blue (MB) served as negative and positive controls, respectively. As with MB, compound **1o** limited parasite multiplication in mice, albeit to a lesser extent (Fig. 3a). Importantly, three doses were sufficient to fully block parasite transmission to mosquitoes (Fig. 3b,c, 4 days post passage). A reduction of parasite transmission to mosquitoes was already visible 24 h after the first treatment (Fig. 3b,c, 2 days post-passage), with reduced infection prevalence and parasite loads in mosquitoes fed on compound **1o**-treated mice compared with control mice.

Parasite gametocytes are the only stage infectious to mosquitoes. To determine whether compound **1o** has any activity on these sexual stages, we exposed *P. falciparum* gametocytes to drug treatment at different times during their maturation. A strong reduction of gametocyte viability was observed for all stages after exposure to low concentrations of compound **1o** (relative survival below 50% for concentrations > 5 nM), although we never detected complete killing of gametocytes, even at concentrations > 1 μM (Fig. 3d). The fact that early stage gametocytes survived better at 100 μM than at 1 μM suggests that at high concentrations, the compound could have a gametostatic activity. Taken together, these data and the activity of compound **1o** against *P. berghei* asexual parasites *ex vivo* suggest that the transmission blocking effect observed above is explained by a combination of the activity of **1o** on gametocytes and a reduced gametocytemia following the impairment of asexual

stages, and/or a decrease in gametocyte commitment on treatment.

**Antimicrobial profile and effects on *Schistosoma mansoni*.** To learn more about specificity and potential application of the steroid compounds, the two best compounds **1o** and **2o** were tested for their antimicrobial spectrum at the Hans Knöll Institute, Jena (Supplementary Tables 3 and 4). The compounds showed moderate activity against selected bacteria and weak activity against fungi. Antimicrobial activity was detectable in 4 out of 11 cases (*Escherichia coli*, *Mycobacterium vaccae*, *Sporobolomyces salmonicolor* and *Candida albicans*); however, micromolar concentrations of the compounds were required. No biologically relevant inhibition was determined for *Bacillus*, *Pseudomonas*, *Staphylococcus*, *Enterococcus* or *Penicillium* (Supplementary Table 3). The antifungal profile against 6 ascomycetes and 2 zygomycetes indicated weak activity only (Supplementary Table 4).

Interestingly, however, the steroid compounds were found to have remarkable physiological and morphological effects on adult *Schistosoma mansoni*, leading to the death of this trematode parasite *in vitro*. To investigate the effects of the compounds **1o**, **2o** and **1c** in more detail, we made use of an established *in vitro* culture for adult *S. mansoni*[20]. The three compounds were tested in concentrations of 1–100 μM each, and their effects were first checked via bright-field microscopy with respect to physiological parameters such as pairing stability,

egg production, and survival. Starting with 1 µM (**1c**, **2o**) and 5 µM (**1o**), respectively, all compounds showed a negative influence on worm physiology. Using 10 µM and more, the most remarkable effects on all three parameters were observed with **2o**, followed by **1c** and **1o**. Using **2o**, reduced motility and viability were observed starting at 1 µM, while 10 µM of this compound caused the death of the parasites within a week (Supplementary Fig. 9). This was accompanied by tegumental invaginations and oedema-like swellings of the body. CLSM indicated an enlarged gut lumen and degradation of the gastrodermis, which led to the accumulation of degraded tissue and aggregate formation. Furthermore, we observed a disorganization of oocytes within the ovary and reduced sizes of testicular lobes in males (see Fig. 4 and detailed explanations in the legend).

All tested compounds affected physiology and morphology in *S. mansoni* that included changes in the female ovary as well as in the gut of both genders, finally leading to the death of the blood fluke. Interesting was the tissue dilatation of the gut associated with the degradation of the gastrodermis and the accumulation of particle aggregates of remarkable size. In this intensity, such a phenotype has not yet been observed with other substances applied to adult schistosomes *in vitro*[20], although it resembled a phenotype previously observed for adults treated with imatinib or its derivatives dasatinib or nilotinib to some extent. However, in none of the latter cases were tegumental invaginations or a tissue dilatation of this intensity observed, nor such a strong accumulation of aggregates. It should further be noted that in comparison with a unicellular organism such as *Plasmodium*, it usually takes much higher drug concentrations (accompanied by lower selectivity indices) to target multicellular organisms such as *Schistosoma*. In the case of the arylmethylamino steroids, lower micromolar concentrations were required to see clear effects on schistosomes. This is exactly in the range of the gold standard praziquantel, which is toxic for schistosomes at 5–10 µM but no longer at 1 µM *in vitro*[21].

**Cytotoxicity and pharmacokinetic parameters**. Selected steroid compounds were tested for their antiproliferative activity in HUVEC cells and K-562 cells with GI$_{50}$-values between 0.1 and 108.5 µM. Cytotoxicity tests were carried out on HeLa

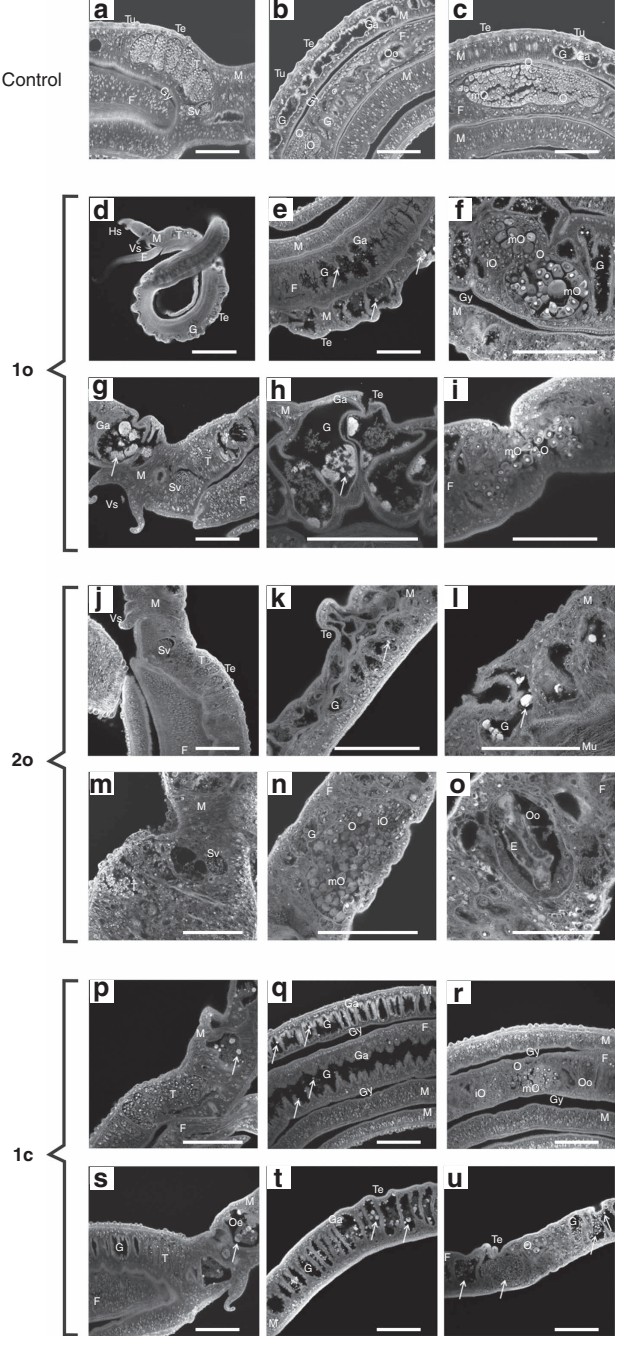

**Figure 4 | Morphological effects on adult *S. mansoni* in vitro.** Compounds **1o**, **2o**, and **1c** (5 µM each) were administered over 6–13 days before effects were investigated via confocal laser scanning microscopy. (**a–c**) Untreated schistosome couples exhibiting a smooth tegumental surface with tubercles. (**a**) The testes are composed of lobes containing spermatogonia and differentiated spermatozoa accumulating in the sperm vesicle. (**b**) The gut lumen is surrounded by the gastrodermis. (**c**) The ovary of the female exhibits a bulb-like structure with mature oocytes at the broader, posterior part and immature oocytes at the narrow, anterior part. (**d,e**) The ootype is the egg-forming organ. (**d–i**) **1o** (9-day treatment); swellings and invaginations occurred at the tegument. Arrows mark aggregates of degradation products of the gastrodermis within the gut lumen. (**f**) The number of mature oocytes was reduced, some occurred within the anterior part of the ovary, which normally only contains immature oocytes. (**g,h**) After 13-day treatment, swellings, invaginations and the size of the aggregates increased. (**i**) No more immature oocytes occurred. (**j–o**) **2o** (5 µM; 6-day treatment); (**j**) the diameter of the testicular lobes was reduced, the ovary was disorganized. (**k,l**; arrows) Oedema-like swellings, gastrodermis degradation, and aggregate formation were visible. (**m**) After 9-day treatment, diameter of the testicular lobes and number of spermatozoa within the seminal vesicle were reduced. (**n**) The morphology of the ovary was disturbed, (**o**) and eggs were deformed. (**p–u**) **1c** (5 µM; 9-day treatment); (**p**) diameter of the testicular lobes and ovary appeared normal, number of mature oocytes was smaller. (**p,q**) Aggregate-like degradation products of the gastrodermis occurred within the gut lumen; size of aggregates was larger in males (**p,q**) than in females (**r**). Gut swelling and tegument invaginations were not obvious. (**s**) After 13-day treatment, the diameter of the testicular lobes was reduced. (**t,u**) Gut swelling and tegument invagination appeared. Morphology of ovary was disturbed and number of mature oocytes reduced. (**s–u**) Degradation aggregates appeared within gut lumen and oesophagus area. After 9 days, precipitates were larger in males (**s,t**) compared with females (**u**). E, egg; F, female; G, gut; Ga, gastrodermis; Gy, gynecophoral canal; Hs, head sucker; M, male; T, testis; Te, tegument; Tu, tubercle: O, ovary; Oe, oesophagus; mO, mature oocytes; iO, immature oocytes; Oo, ootype; Sv, seminal vesicle; Vs, ventral sucker. Scale bars, 100 µm, except **d** (400 µm) and **o** (50 µm).

cells, resulting in $CC_{50}$-values between 5.9 and $>128\,\mu M$, which is 2–4 orders of magnitude higher than the $IC_{50}$ values determined against malaria parasites (Table 1). In addition, cytotoxicity tests with compound **1o** were carried out on immortalized human hepatocytes, Fa2N-4 cells, resulting in an $LC_{50} > 50\,\mu M$. The best compound **1o** has furthermore been tested in an acute toxicity *in vivo* model at the Swiss Tropical Institute, Basel. The compound was applied at an accumulative dose of $100\,mg\,kg^{-1}$ (p.o. as well as i.p.), but no acute toxicity symptoms were observed.

First pharmacokinetic data for the best compound **1o** were collected at the Sanford Burnham Prebys Medical Discovery Institute (Supplementary Table 6). Aqueous solubility, plasma stability, plasma protein binding, hepatic microsome stability, and membrane permeability using the parallel artificial membrane permeability assay (PAMPA) were determined. Although compound **1o** was found to have rather poor aqueous solubility ($<1\,\mu g\,ml^{-1}$ across a range of pH), it displayed good stability in plasma (62 and 58% remaining, respectively, after a 3 h incubation with human and mouse plasma) and moderate microsomal stability ($\sim38\%$ remaining after a 1 h incubation with human or mouse liver microsomes).

**Influence of metal or haem binding and catalysis**. To gain more insight into the mechanism of action of the steroid compounds, the binding capacities of the most potent compounds **1o** and **2o** towards Cu(II) and Fe(III) were first evaluated under quasi-physiological experimental conditions (pH 7.5). Both compounds were shown to lead to 1:1 stoichiometric complexes with stability constants log $K_{LM}$ (L = **1o** or **2o**, M = Cu(II) or Fe(III)) ranging from = 4.1 to 4.7 (Supplementary Note 3, Supplementary Fig. 17, and Supplementary Methods). Compounds **1o** and **2o** share the same bidentate N,O binding site suitable for speciation of soft divalent or hard trivalent metal ions.

Interestingly, the most potent antimalarial compounds (**1o** and analogues found in the present work) are the ortho-substituted derivatives, suggesting that generation of radicals is the most probable explanation for the increased efficacy of the chemical series. Similar *ortho*-substituted ligands of transient metal complexes, such as the reduced benzoylmenadione metabolite of the antimalarial plasmodione, which presents an oxygen-rich bidentate site suitable for Fe(III) chelation[22], were reported to become redox-inactive when their effects were antagonized with the known iron(III) chelator desferoxamine (DFO). In this study a most potent antagonistic effect (sum $FIC_{50} = 2.4$) induced by DFO was observed when combined with plasmodione in cell culture assays using RBCs parasitized by *P. falciparum* 3D7.

In parallel, the ability of a broader range of compounds (**1o**, **2o**, **1a** and **1h**, used as a negative control, see Fig. 1) to interact with haem was evaluated (Supplementary Note 3, Supplementary Table 7, Supplementary Figs 10–16 and Supplementary Methods). Regardless of the nature of the compound, strong interactions ($K_D$ in the $\mu M$ range) with the haem dimer were measured (that is, haematin predominates as a $\pi$-$\pi$ dimer in solution at pH 7.5) and compare well with chloroquine **CQ**, a well-known antiparasitic drug that acts as a haemozoin inhibitor. $\pi$-$\pi$-interactions and hydrophobic interactions are likely the driving forces accounting for such strong interactions. It therefore appears that **1o** and **2o** displaying an N,O bidentate site are capable of interacting with electroactive metal ions such as Cu(II) or Fe(III). With respect to haem ligation, the steroid unit was demonstrated to be of great importance. The capacities of these systems to prevent haemozoin

formation were then investigated using the ESI-MS collision-induced dissociation method. Interestingly, **1o** ($DV_{50} = 289\,V$, bottom plateau of 20% at 400 V) and **2o** ($DV_{50} = 280\,V$, bottom plateau of 21% at 400 V) were found to be the most potent compounds to prevent haemozoin crystallization and are better than the well-known antiparasitic drugs chloroquine ($DV_{50} = 201\,V$, bottom plateau of 0% at 400 V) or amodiaquine ($DV_{50} = 212\,V$, bottom plateau of 7% at 400 V). The significant residual amount of haem **1o** or haem **2o** adducts at a high fragmentor voltage of 400 V is indicative of covalent binding, likely through quinone methide formation[23] and cross-linking to haem. The 2-hydroxyarylmethylamino moiety was further demonstrated to be of crucial importance since model compounds **1a** and **2a**, which are lacking such an electroactive unit, are less efficient. The covalent character of the haem adducts observed with **1o** and **2o** was further substantiated by co-incubation of the compounds and haem at pH < 1 followed by ESI-MS.

In conclusion, **1o** and **2o** bearing a 2-hydroxyarylmethylamino moiety are potent metal (copper or iron) chelators and likely covalently associate with haematin via quinone methide formation and subsequent cross-linking. No significant difference has been found between **1o** and **2o**, showing that steroid conformation is not of crucial importance. However, the presence of the steroid moiety has been clearly demonstrated to favour the physiochemical properties towards inhibition of haemozoin formation, leading to increased antimalarial activities of both lead agents. On the basis of these data and taking into account the fact that parasitized red blood cells contain high amounts of haem and free iron, it can be estimated that more than 80% of compound **1o** is present in complexed form within the infected RBC (Supplementary Note 4 and Supplementary Fig. 18).

## Discussion

We identified and characterized arylmethylamino steroids as a class of compounds with remarkable activity against malaria parasites and schistosomes. Transmission-blocking activity, oral availability, and low cytotoxicity further support their potential.

The presently synthesized and available compound pool comprises about 60 steroid derivatives and a series of non-steroidal analogues that can be employed for SAR studies and considerations concerning a lead structure for further optimization (Fig. 5). Present SAR data indicate that the hydrophobic steroid component and a hydroxyarylmethylamino group are essential for the antimalarial action of the compounds. This gives rise to some key considerations (Fig. 5): the delivery of small molecules to living cells is typically mediated by polymers, lipids, sophisticated nano-sized drug delivery systems[24], or by altering membrane permeability, for example, via covalent conjugation of target molecules with cholane derivatives[25]. In our context, the lipophilic steroid moiety is considered to mediate cellular uptake and intracellular transport processes. Steroid conjugates often readily enter living cells, either via membrane association and/or endocytotic pathways as known from cholane derivatives[26] or specifically via steroid binding sites as known from steroid hormones[27]. In our study, the highest biological activity was found among estratriene derivatives (series **1–11**), while cholane derivatives (series **12–14**) were less efficient. *De facto*, none of the non-steroid-derived analogs (series **15–20**) exhibited antimalarial activity comparable to the steroid compounds. The $IC_{50}$ values on *P. falciparum* 3D7 asexual blood stages were about $2\,\mu M$ for **17c**, $4\,\mu M$ for **18c**, $5.6\,\mu M$ for **20c**, $>6\,\mu M$ for **15c** and **16c**, and $>15\,\mu M$ for **19c**. Notably, **18c** is derived from the lipophilic 1-adamantylamine,

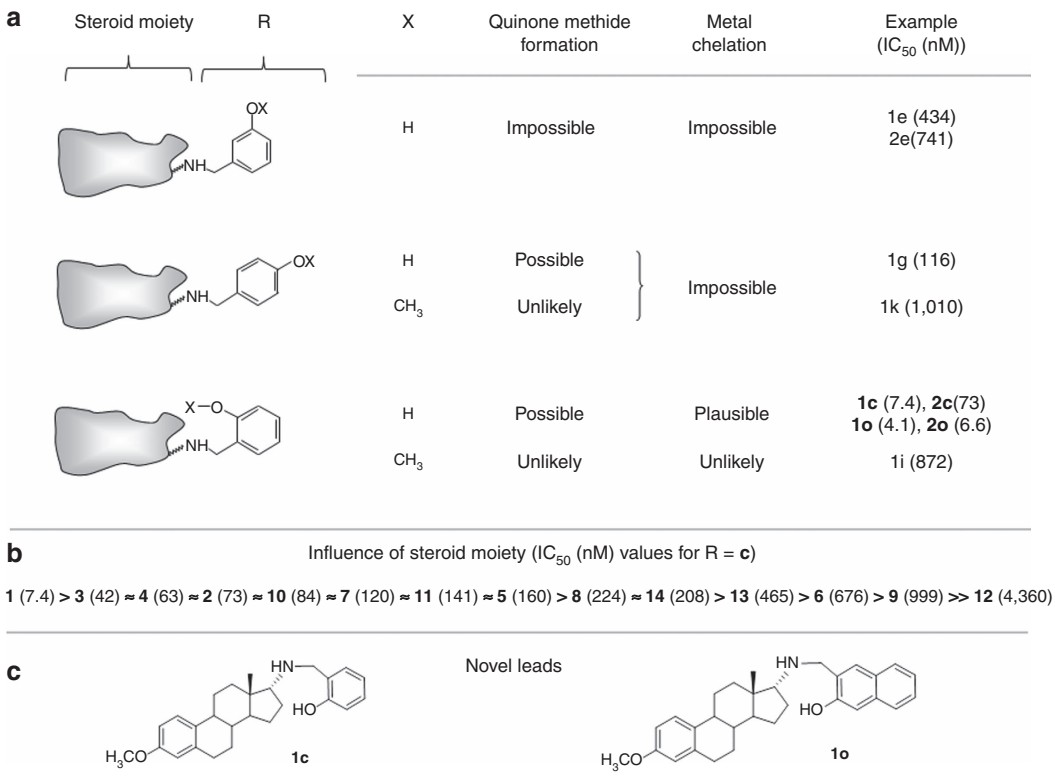

**Figure 5 | Working hypothesis for the mode of action and optimization of R. (a)** *Ortho*-hydroxylated arylmethyl amines exhibit the highest antimalarial activity. A chelate involving quinone methide mechanism is substantiated via structure/activity considerations. Both the redox pair phenol/quinone and the highly reactive quinone methide itself may be causative for biological activity. **(b,c)** The steroid core also plays a crucial role. The highest activity is found for estratriene derivatives.

which is closely related to the neuroprotective agent *memantine* (3,5-dimethyl-1-adamantylamine). On the basis of these data, the steroid part is essential for the biological activity of the compounds. This interesting new chemical entity seems to vectorize the aminocresols to parasitized red blood cells and is potentially recognized by transporters. The present study might therefore initiate new investigations based on the use of steroids to target not only drugs, but also fluorophores or other chemical probes, to cells of interest. On the basis of the properties of our compounds, the sex hormone-binding globulin (SHBG), the principal carrier of oestrogens and androgens in the blood of almost all vertebrate species[28], may be involved in drug transport in the *in vivo* experiments. Furthermore, in our approach, a lipophilic moiety joined to a hydrophilic one could result in an inhibitor that might destabilize cell membranes as known from phosphatidylcholine inhibitors[29,30].

Notably, the 2-hydroxyarylmethylamino-group is present in all of our best-acting derivatives. Such compounds are well known for forming *ortho*-quinone methides of high inherent reactivity[29,31,32]. There is some evidence that quinone methide intermediates, which may alter or inhibit the action of biomolecules via covalent cross-linking, play a key role (Fig. 5). In our approach, methylation of the phenolic group (R = **i** and **k**) or movement of the OH function to the meta-position (R = **e**) prevents the formation of quinone methides. Even these compounds exhibit a clearly decreased antimalarial activity. Movement of the phenolic group to the para-position recovers biological activity. Introduction of bulky *tert*-butyl groups into *ortho*- and *para*-positions prevents cross-linking capability (**6w**). All these findings make a quinone methide mechanism

permitted by R very likely. Furthermore, *ortho* phenols are most efficient, while their methyl ethers as well as *meta*-substituted phenols are less active (Fig. 5). This fact supports the participation of a chelation mechanism, including activation processes and ongoing redox processes mediated by metals. The recovery of antimalarial activity from *meta*- to *para*-substituted phenols, however, suggests that additional mechanisms are likely to support activity, as the latter cannot act as bidentate ligands. Given the lipophilicity of our steroid compounds as well as their basic nature, it is furthermore most likely that they access the food vacuole of the parasite. At the pH of the organelle (5.0), these molecules will be protonated and concentrated. Protonation of an aminocresol will increase the potential for a site-specific formation of a reactive species, limiting any safety risk. Furthermore, the SAR demonstrates that compounds not having an aminocresol retain potency (albeit reduced) suggesting that the structures retain other pharmacology unrelated to this covalent binding; this is an innovation—delivering a compound having polypharmacology—with the benefits such as resistance selection apparent.

When proposing steroid-based compounds as anti-infective agents, a potential intrinsic biological activity of the steroid moiety determined by its steric and electronic fit with steroid binding sites should be considered. In oestrogens, hormone action is mainly determined by the specific interaction of the 3- and 17β-OH functions. Both binding sites are impaired in series **1–8** and **10–11** by 3-OH methylation and in series **1–6** additionally by the lack of the 17β-OH function. Therefore, potential side effects based on steroid hormone activity are unlikely to occur, particularly taking the short-term course of an antimalarial therapy into account. Furthermore, in

our experiments no initial liabilities were identified at high µM concentrations *in vitro* and at 100 mg kg$^{-1}$ *in vivo* in *P. berghei*-infected mice. However, more extensive and thorough testing will be necessary during further preclinical development of one of the two best compounds or optimized analogues.

Finally, phenoxide binding to haem iron[33] might contribute to the antimalarial effects of our compounds where the 2-hydroxyarylamino moiety may chelate or associate with metals. For derivatives with this configuration, an increased antimalarial activity was found compared with the 4-isomer, which may also furnish quinone methide intermediates but does not chelate metals. Utilizing metal catalytic effects as a driving force and their dependence on substitution patterns of quinone methide precursors have been described in other contexts[34] and were studied in detail for our antimalarial compounds (see above).

The most potent steroid compounds discovered here actually represent a subclass of aminocresols, a group of antimalarials that have shown up in various antimalarial studies since the 1940s and include amodiaquine. However, none of these compounds has so far had a steroid attached. A prominent example of aminocresols is WR194965 (ref. 35). This compound has strong antimalarial activity against *P. falciparum* and *P. vivax* and was even taken into human trials. However, since mefloquine was discovered at the same time, WR194965 was not followed up in detail. This is also the reason why only limited toxicity data are available. A second example is MK4815 (from Merck), which was also identified via high-throughput screening and was found to be orally active in a *P. berghei* mouse model, even when applied as a single dose[36]. However, the IC$_{50}$ against *P. falciparum* 3D7 was about one order of magnitude higher than that of our steroid-based compounds. Indeed there are additional theoretical concerns that exist for aminocresols in general. First, there is the potential for unwanted toxicity associated to quinone methide formation. Second, depending on the lipophilicity and the basicity of the compounds, there is potential for cardiovascular and CNS safety issues. All of these would need to be carefully monitored in preclinical safety studies before starting human studies. However, even in reasonably detailed toxicology studies, the respective compounds, including our steroid derivatives, have hardly shown any toxicity. Furthermore, as indicated by initial studies in cell culture using a genetically encoded redox probe (Supplementary Fig. 8), the redox stress induced by the steroid compounds might contribute to their mechanism of action but is unlikely to represent a major risk for patients with glucose-6-phosphate dehydrogenase deficiency. Interestingly, Jacobus Pharmaceuticals has recently delivered a second generation WR194965 named JPC3210 with reported excellent efficacy and half-life[37]. Much has already been said about pan-assay interference compounds (PAINs) that cannot be developed into selective covalently binding drugs[38,39]. However, not all compounds with PAIN characteristics should be discarded *a priori* as antimicrobial agents because the compounds are expected to be administered for rather short-term treatment compared with drugs acting in other fields such as the treatment of cancer or cardiovascular diseases. Although the authors of course fully agree that the potential safety issues of PAINs need to be taken into account thoroughly, it is a big challenge to fight dogmas and taboos in the field of anti-infective drug discovery. Furthermore, even with PAINS characteristics, compounds with excellent potencies should be considered for chemical probe discovery[40] to help and improve our understanding of drug targets/transporters and pathways.

The next steps for further optimizing our steroid compounds include in-depth analyses of pharmacokinetic and toxicological properties, further improvement of lipophilicity and ADME properties including oral bioavailability and enhancement of potency *in vivo*. One starting point could be the further optimization of R, for example, by introducing suitable leaving groups. Furthermore, the potential of combining the steroid compounds with known antimalarial agents should be systematically evaluated. To reduce potential hormone activity a basic nitrogen function has already been introduced into the compounds. Further approaches could be taken including the preparation of analogues with altered substitution patterns and altered stereochemistry at the gonane core and the introduction of a second (or third) hydroxyarylmethylamino group. A more detailed discussion of lead optimization is given in Supplementary Note 1. On the basis of our studies and the corresponding considerations we propose here a novel approach to drug development for fighting malaria, schistosomiasis and potentially other parasitic diseases.

## Methods

**Cultivation of *Plasmodium falciparum*.** The chloroquine-sensitive *P. falciparum* strain 3D7 Netherlands and the chloroquine-resistant strain Dd2 (kindly provided by Michael Lanzer, Heidelberg in 2000) were grown in continuous culture as described by Trager and Jensen[41] with slight modifications[42]. Unless otherwise stated, parasites were maintained at 1 to 10% parasitaemia and 3.3% haematocrit in an RPMI 1640 culture medium supplemented with A+ erythrocytes (purchased from the blood bank of the University Hospital Giessen and Marburg, UKGM), 0.5% lipid-rich bovine serum albumin (Albumax), 9 mM (0.16%) glucose, 0.2 mM hypoxanthine, 2.1 mM L-glutamine, and 22 µg ml$^{-1}$ gentamicin. All incubations were carried out at 37 °C in 3% O$_2$, 3% CO$_2$, and 94% N$_2$ unless otherwise stated. Synchronization of parasites in culture to ring stages as a starting population was carried out via treatment with 5% (w/v) sorbitol. The parasites were used for the experiments delineated below.

**IC$_{50}$ values on *P. falciparum in vitro* and drug combinations.** Isotopic drug sensitivity assays were employed as described[41] to investigate the susceptibility of *P. falciparum* to the steroid compounds. The procedure depends on the incorporation of radioactive $^3$H-hypoxanthine, which is taken up by the parasite as a precursor of purine deoxynucleotides for DNA synthesis[43]. In 96-well microtitre plates (Nunc$^R$), a twofold serial dilution of the starting concentration of each pharmacologically active compound to be tested was carried out. Parasites were incubated at a parasitaemia of 0.25% (>70% ring forms) and 1.25% haematocrit in hypoxanthine-free medium. After 48 h, 0.5 µCi $^3$H-hypoxanthine was added to each well, and the plates were incubated for another 24 h. The cells of each well were then collected on a glass fibre filter (Perkin-Elmer, Rodgau-Jügesheim, Germany), washed, and dried. Their radioactivity in counts per minute was considered to be proportional to the number of parasites in the well. All IC$_{50}$ values were determined at least in quadruple. IC$_{50}$ values (drug concentrations that produce 50% reduction in the uptake of $^3$H-hypoxanthine) were calculated.

**IC$_{50}$ values on *P. falciparum* gametocytes *in vitro*.** Sensitivity of *P falciparum* gametocytes to lead compound **1o** was determined by measuring the activity of luciferase in luciferase-expressing gametocytes after an exposure of 72 h to different concentrations of compound **1o**.

The pfs16-GFP-Luc (NF54 background) parasite strain[44] was maintained at 1 to 10% parasitaemia in an RPMI 1640 culture medium supplemented with A+ erythrocytes (2% haematocrit), 10% decomplemented human serum, 9 mM (0.16%) glucose, 0.2 mM hypoxanthine, 2.1 mM L-glutamine, and 20 µg ml$^{-1}$ gentamicin. Parasite cultures were incubated at 37 °C in 5% O$_2$, 5% CO$_2$, and 90% N$_2$. Asexual blood stage parasites were first synchronized by 5% D-sorbitol (wt vol$^{-1}$) treatment for 10 min at 37 °C for two or more successive growth cycles. Synchronous parasites at 3% haematocrit were cultured in 10 cm Petri dishes to 10% parasitaemia (ring stage), at which point gametocytogenesis was induced by feeding cultures with a mixture of conditioned and fresh media (1:1). The next day, trophozoite cultures were diluted fourfold. N-acetyl glucosamine (NAG; Sigma-Aldrich) was added to a final concentration of 50 mM after all schizonts had ruptured. NAG treatment was maintained for the next two to three cycles of reinvasion to eliminate all asexual forms from the culture. Immature gametocyte stages (I and II) were purified on a Percoll gradient[45] and incubated directly in 96-well plates at 9%, 12%, or 45% gametocytaemia depending on the experiment (150 µl well$^{-1}$). Gametocyte cultures were exposed for 72 h to dilutions of compound **1o**, methylene blue (MB) (Sigma M9140) or DMSO (0.5%) starting at day 3, 8, or 11 after gametogenesis induction. For this, compound **1o** (10 mM) and MB (10 mM) were dissolved in 100% DMSO and H$_2$O, respectively, diluted 1:100 in RPMI, and further diluted in RPMI to the proper concentrations. 150 µl of these solutions were added to the

gametocyte culture in triplicate. Following drug exposure, parasites were cultivated for two additional days in normal medium before centrifugation and freezing at $-80\,^{\circ}C$, and luciferase activity was measured.

**Luciferase assays in *P. berghei* and *P. falciparum* in vitro.** Plates frozen at $-80\,^{\circ}C$ were thawed at $37\,^{\circ}C$. The cells were lysed at room temperature in $10\,\mu l$ 1x luciferin lysis buffer (Luciferase Cell Culture Lysis $5\times$ reagent, Promega), and $100\,\mu l$ luciferase assay substrate was added to each well (Luciferase Assay System, Promega). Luciferase activity was measured on a luminescence plate reader (Mithras LB940 luminometer, Berthold Technologies). The mean of the luciferase activity was calculated for each technical triplicate and normalized as follows: $(Luc_x - Luc_{RBC})/(Luc_{solvent} - Luc_{RBC})$ with $Luc_x$ and $Luc_{solvent}$ = mean luciferase activity for parasites incubated with compound x and with solvent (DMSO for **1o** and PBS for MB), respectively, and $Luc_{RBC}$ = mean luciferase activity of non-infected red blood cells (background). $IC_{50}$ values were calculated using Prism (GraphPad, log(inhibitor) versus normalized response—variable slope) in three independent experiments for the *P. falciparum* gametocyte assay and five for the *P. berghei* ex vivo drug assay.

**In vivo testing in the *P. berghei* mouse model.** The animal experiments described here were performed at the Swiss Tropical and Public Health Institute (Swiss TPH, Basel, Switzerland), they were approved by the Swiss Cantonal Authorities ('Kantonales Veterinäramt Basel Stadt') adhering to local and national regulations of laboratory animal welfare in Switzerland. The best compounds were tested in the murine *P. berghei* model essentially as described[17–19]. The infection was initiated at day 0 with the *P. berghei* GFP ANKA malaria strain (donation from AP Waters and CJ Janse, Leiden University). From donor mice (female NMRI mice, 18–20 grams (about 3 weeks) from Charles River Germany) with ~30% parasitaemia, heparinized blood was taken and diluted in physiological saline to $10^8$ parasitized erythrocytes/ml. An aliquot (0.2 ml) of this suspension was injected intravenously into experimental and control groups of mice (female NMRI mice, 18–20 grams (about 3 weeks) from Charles River Germany). Usually, in untreated control mice parasitaemia rose regularly to ~30% by day 3 post-infection. Control mice died between days 6 and 7 post infection. In the experiments described here, however, control animals ($n = 5$) were killed on day 4 post infection for ethical reasons.

The steroid compounds were prepared at an appropriate concentration as a solution/suspension in Tween 80/alcohol (7:3, Tween 80 and absolute ethanol, respectively), followed by a 10x dilution in water. They were administered to groups of three mice in three doses (24, 48 and 72 h post infection) or four doses (4, 24, 48 and 72 h post infection). The route of administration was i.p., p.o., s.c. in a volume of $10\,ml\,kg^{-1}$.

The degree of infection (parasitaemia expressed in % of infected erythrocytes) was determined by FACS analysis on day 4 (96 h post infection). The difference of the mean infection rate of the control group ($= 100\%$) to the test group was calculated and expressed as a percent reduction. For example, activity determination with a mean of 2% parasitaemia in treated mice and a mean of 40% parasitaemia in the control animals is calculated as follows: $(40 - 2\%)/40\% \times 100 = 95\%$ activity. The survival time in days was recorded up to 30 days after infection. A compound was considered curative if the animal survived to day 30 after infection with no detectable parasites (confirmed by light microscopy).

**In vivo maintenance of *P. berghei* parasites.** The animal experiments described in this part were performed at the *Institut de Biologie Moléculaire et Cellulaire* (Strasbourg, France), using facilities and protocols adhering to national regulations of laboratory animal welfare in France. Facilities and protocols have been certified by the regional veterinary services (authorization N° F67-482-2) and by the national ethics committee in animal experimentation (authorization N° 04480.02), respectively. 8- to 12-week-old male Hsd:ICR mice (bred in-house from couples purchased from Envigo) were infected with the *P. berghei* ANKA malaria strain constitutively expressing GFP or GFP luciferase[46,47]. For this, blood was taken by heart puncture from a donor mouse with a parasitaemia of 3–5%, diluted in PBS to $10^8$ parasitized erythrocytes per ml. 0.2 ml of this suspension was injected intravenously into mice. Parasitaemia was monitored via FACS analysis (FACSCalibur, BD Bioscience). The *P. berghei* parasites maintained in vivo were then used for the ex vivo drug assay and transmission-blocking assays.

**Determination of $IC_{50}$ values on *P. berghei* parasites ex vivo.** Sensitivity of *P. berghei* asexual stages to lead compound **1o** was determined by measuring the activity of luciferase in a parasite expressing constitutively GFP-Luciferase after an exposure of 24 h to different concentrations of compound **1o** in ex vivo conditions[46]. Briefly, infected blood was collected by heart-puncture from a mouse with parasitaemia between 1 and 3%. Blood was washed twice in RPMI 1640 culture medium supplemented with fetal calf serum (FCS) at 25% and resuspended in the same medium. Infected blood samples were exposed for 24 h to dilutions of compound **1o** and methylene blue (Sigma) (0, 1, 3, 10, 30, 100, and 300 nM) or DMSO (0.2%). For this, compound **1o** (10 mM) and MB (10 mM) were dissolved in 100% DMSO and $H_2O$, respectively, diluted 1:100 in RPMI, and

further diluted in RPMI to the proper concentrations. $50\,\mu l$ of these solutions were added to each well containing $50\,\mu l$ of infected blood (final haematocrit 2%) in triplicate. The plate was incubated for 24 h at $37\,^{\circ}C$ in 5% $O_2$, 5% $CO_2$ and 90% $N_2$; supernatant was removed before freezing at $-80\,^{\circ}C$; and luciferase activity was measured.

**In vivo transmission-blocking assay.** *Anopheles gambiae* mosquitoes (G3 strain, donation from the Kafatos laboratory at EMBL in 2002) were bred following standard procedures[48].

Compound **1o** was prepared in solution/suspension in Tween 80/absolute alcohol (7:3), followed by a 10x dilution in water to reach the final concentration of $10\,mg\,ml^{-1}$. ProveBlue (Inresa Pharma), a heavy metal-free solution of MB ($5\,mg\,ml^{-1}$) was used as a positive control. Both solutions were injected i.p. in infected mice as daily doses (24, 48 and 72 h post passage) ($100\,mg\,kg^{-1}$ for **1o** and $15\,mg\,kg^{-1}$ for MB). Negative control mice were i.p. injected with an equivalent quantity of solvent.

Parasitaemia was determined daily via FACS analysis (FACSCalibur, BD Bioscience). In parallel, groups of 40 *Anopheles gambiae* mosquitoes (4–7 days old) were allowed to take a blood meal on treated and control infected mice (see paragraph '*In vivo* maintenance of *P. berghei* parasites for ex vivo drug assay and transmission-blocking assay') at 48, 72, 96 and 120 h post infection (one group of 40 mosquitoes per mouse per day). For this, mice were anaesthetized via an i.p. injection of 5% Rompun (Xylazine) and 10% Imalgène (Ketamine) diluted in saline solution ($100\,\mu l$ per $10\,g$) and exposed to mosquito bites for 15 min. Fully engorged mosquitoes were sorted, and their midgut was dissected in $1\times$ PBS at 7 days post infection. Each midgut was imaged using a Nikon fluorescence microscope AZ100, and fluorescent parasites (oocyst stage) were counted using the Image J watershed program. The prevalence (percentage of infected mosquitoes) and infection level (average number of parasites per midgut in mosquitoes carrying at least one parasite) were plotted using Prism. Three independent experiments were performed, and significance for differences between treatments was calculated using generalized linear models with treatment, time and repeats as variables using the JMP software (SAS). Data were modelled with normal distributions and an identity link for parasitemia, a logit link for prevalence, and a log link for the level of infection.

**Inhibition tests on bacteria and yeast.** The following organisms were tested at the Hans Knöll Institute in Jena, for their susceptibility to the compounds **1o** and **2o**: *Bacillus subtilis* JMRC:STI:10880, *Staphylococcus aureus* JMRC:ST:10760, *Escherichia coli* JMRC:ST:33699, *Pseudomonas aeruginosa* JMRC:ST:33772, *Pseudomonas aeruginosa* JMRC:ST:337721, *Staphylococcus aureus* JMRC:ST:33793 (multi-resistant), *Enterococcus faecalis* JMRC:ST:33700 (Vancomycin-resistant), *Mycobacterium vaccae* JMRC:STI:10670, *Sporobolomyces salmonicolor* (*Basidiomycetes* yeast) JMRC:ST:35974, *Candida albicans* (*Ascomycetes* yeast) JMRC:STI:25000, *Penicillium notatum* JMRC:STI:50164.

The bacteria were propagated in Mueller Hinton Broth (Oxoid)[49]. Yeast was propagated in YPD (yeast extract peptone dextrose): 1% yeast extract, 10 g; 2% peptone, 20 g; 2% agar, 20 g; 2% dextrose (glucose), 20 g; Sigma-Y1375. All bacteria and yeast suspensions were freshly made. The cultures were incubated at $37\,^{\circ}C$ for 16 h. Afterwards, the bacteria concentration from each culture was determined, and 34 ml of nutrient agar was loaded with $10^7$ cells. These cultures were stored at 6-8 °C and could be used for seven days for the preparation of test plates. 34 ml of nutrient agar was liquefied and inoculated with the calculated amount of inoculum bacteria suspension at 48–50 °C, a temperature suitable for vegetative forms of bacteria. The inoculated nutrient media (34 ml each) were immediately poured into the prepared test plates (3 mm layer). With a punching device, 12 holes per test plate were punched out. For inhibiting bacterial growth, ciprofloxacin was used as a reference substance. The test substances were dissolved in distilled water, and $5\,\mu g\,ml^{-1}$ and $50\,\mu l$ was added to each bacterial culture. For inhibition of fungal growth, amphotericin B served as the reference substance. The test substances were diluted to $10\,\mu g\,ml^{-1}$ in DMSO/methanol; $50\,\mu l$ was added to each culture.

**Bacterial growth inhibition assay.** Configuring the test plate; volumes/punched hole: $50\,\mu l$; test solutions/test plate: $1 \times 50\,\mu l$ reference substance. Test plates were cultivated for 18 h at $37\,^{\circ}C$. Analysis: Reading the inhibition zones (IZ): As IZ, the zone in which no growth can be determined with the naked eye is measured; the tiniest colonies on the edge of the inhibition zone were therefore not taken into consideration. The diameter ($\phi$) of the IZ is measured in mm. The analysis of IZ $\phi$ is carried out internationally with the help of statistic procedures, especially regression analysis. With each test approach, the reference substance was brought along as a control. If holes per test plate were deviations from allowed IZ $\phi$, the test was repeated. An analysis at $50\,\mu g$ test substance/diffusion zone showed the following: $-1$ = attempt not measurable; $0$ = no effect; IZ $\phi$ 15–20 mm = 1 moderate effect; 21–25 mm = 2 good effect; > 25 mm = 3 very good effect.

The inhibition of bacterial growth was determined using bacterial cultures grown in Müller-Hinton Bouillon, which was configured at barely visible opacity with the McFarland Standard Nr. 0.5, which indicates a bacterial density of

$10^8$ ml$^{-1}$. This bacterial suspension was diluted with Müller-Hinton Bouillon using 2-fold dilution series in the agar diffusion test.

Analysis: The growth in the inhibition zones around the holes containing active substances is thereby compared with the growth in the growth control (no active substance). Bacterial growth was determined according to four measurements: p = presence of colonies in the inhibition zone, P = presence of many colonies in the inhibition zone, A = indication of inhibition, F = facilitation.

**Spore inhibition test.** The following test organisms were used for their susceptibility to compounds 1o and 2o: *Alternaria alternata* JMRC:SF:09317, *Arthroderma benhamiae* JMRC:ST:35888, *Aspergillus fumigatus* JMRC:Afum:00073 = ATCC 46645, *Aspergillus terreus* JMRC:SF:06307, *Candida albicans* JMRC:ST:35864, *Lichtheimia corymbifera* JMRC:SF:09682, *Penicillium chrysogenum* JMRC:SF:10137, *Rhizopus arrhizus* (syn. *R. oryzae*) JMRC:SF:05857 = ATCC 56536, CBS 112.07.

Fungi were grown on 3% malt extract solidified medium for 1 to 2 weeks at room temperature. Then the plates were washed with 7 ml sterile distilled water, and the spore suspension was filtered through 4 layers of sterile Miracloth (Kalbiochem, USA) to remove the hyphae. The concentration of the spore suspension was set to $2 \times 10^4$ conidia ml$^{-1}$ for the *in vitro* assays.

The spore inhibition assay took place in 96-well microplates (flat base). 50 µl of the prepared spore suspension was seeded and then mixed with 50 µl of the respective inhibitor substance. Tests were started with end concentrations of 1 mg ml$^{-1}$ and were then reduced in a minimum of 10 halving steps. Thus a 2 mg ml$^{-1}$ stock solution diluted in half ten times with solvent (distilled water, 2% DMSO in distilled water, or 2% DMF in distilled water) and each 50 µl was transferred to 50 µl spore suspension. For each test substance, 12 holes on the microtiter plate were filled. After 24 h incubation (25 °C), the germination of the spores was macroscopically and microscopically examined directly in the microtiter plate *via* measurement of the optical density and light microscopy using an inverse stereomicroscope with subsequent photodocumentation, respectively. If necessary, the plates were further incubated for another 24 or 48 h, depending on the extent of fungal germination. The germination was completely inhibited if no mycelium development occurred.

**Schistosome *in vitro* culture including inhibitor treatment and subsequent morphological analysis.** To maintain the life-cycle of *S. mansoni* (in-house Liberian strain)[50], *Biomphalaria glabrata* was used as an intermediate snail host and the Syrian gold hamster as the final host (*Mesocricetus auratus*; males and females; 8 weeks old at the time of infection; purchased from Janvier, France). Adult worms were recovered from hamsters via hepatoportal perfusion 49 days post infection using M199 medium (Gibco) as described previously[20,50]. Experiments with hamsters were done in accordance with the European Convention for the Protection of Vertebrate Animals Used for Experimental and Other Scientific Purposes (ETS No 123; revised Appendix A) and had been approved by the Regional Council (Regierungspräsidium) Giessen (V54-19 c 20/15 c GI 18/10).

Following perfusion, worm couples were collected using featherweight tweezers and washed with M199 medium before keeping in culture in M199 supplemented with FCS (Gibco; 10%), HEPES (Sigma; 1M, 1%), and antibiotic/antimycotic mixture (Sigma; 1%) at 37 °C and 5% CO$_2$ (5). Inhibitor treatment was performed up to 14 days with 1–100 µM of each of the compounds 1c or 1o. Control couples were kept in culture for the same time periods without adding the compound, but otherwise they were treated using the same conditions. During the treatment periods, pairing stability, egg production, and vitality were monitored daily. Pairing stability of schistosome pairs was approved when males kept their female partners within the gynecophoral canal while being sucked with their ventral suckers to the Petri dish. Egg production was determined via bright-field microscopy, counting eggs each day. Separation of couples and detaching from the Petri dish and/or if worms were lying on their sides were evaluated as signs of decreasing vitality. Absence of motility and/or gut peristalsis were judged as signs of death.

Selected samples were morphologically analysed via confocal laser scanning microscopy (CLSM) using a Leica TSC SP2 microscope, a 488 nm He/Ne laser, and a 470 nm long-pass filter in reflection mode as described earlier (3, 5). In short, worm couples were prepared via fixation for at least 24 h in AFA (ethanol 95%, formaldehyde 3%, and glacial acetic acid 2%), stained for 30 min with 2.5% hydrochloric carmine (Certistain, Merck), and destained in acidic 70% ethanol. After dehydration for 5 minutes in 70, 90 and 100% ethanol, worms were preserved as whole-mounts in Canada balsam (Merck) on glass slides.

**Cytotoxicity and antiproliferative effects on mammalian cells.** Compounds were tested for their antiproliferative activity on HUVEC and K-562 cells and for cytotoxicity on HeLa cells using standard protocols. K-562 (DSM ACC 10), HUVEC (ATCC CRL-1730), and HeLa (DSM ACC 57) cells were cultured as described previously[51]. To assay the antiproliferative and cytotoxic effect, procedures were carried out as described before[12]. In brief, for each experiment with K-562 and HeLa ~10,000 cells were seeded per well of the 96-well microplates. For the cytotoxicity assay, the HeLa cells were preincubated for 48 h without the test substances. Cells were incubated for 72 h at 37 °C in

a humidified atmosphere and 5% CO$_2$. Suspension cultures of K-562 in microplates were analysed by an electronic cell analyzer system CASY 1 (SCHÄRFE, Reutlingen, Germany). The monolayer of the HUVEC and HeLa cells were fixed by glutaraldehyde (MERCK) and stained with a 0.05% solution of methylene blue (SERVA). After gently washing, the stain was eluted by 0.2 ml of 0.33 N HCl in the wells. The optical densities were measured at 660 nm in a microplate reader. The calculations of the different values of GI$_{50}$ and CC$_{50}$ were performed with the software Magellan (TECAN).

Furthermore, cytotoxicity tests were carried out on immortalized human hepatocytes, Fa2N-4 cells (XenoTech, cat.no. IFH15), at the Sanford-Burnham Medical Research Institute. Cells were incubated with a range of concentrations (0.01–50 µM) of compound 1o for 24 h. Cell viability was determined with cellular ATP levels by using the Luminescence ATP Detection Assay System (ATPlite 1 step).

**Pharmacokinetics.** Pharmacokinetic data were obtained for compound 1o at the Sanford-Burnham Medical Research Institute according to established procedures. Compound solubility in an aqueous solution was measured using an automated kinetic solubility method. The concentration of the compound in a saturated pH-buffered aqueous solution was determined via UV absorbance (250–498 nm) and compared with the spectra of a precipitation-free reference solution. The plasma stability of the compound was determined at a single concentration using species-specific plasma. The percentage of parent compound remaining after 3 h was determined via LC-MS/MS. The percentage of compound 1o bound to plasma proteins was determined through rapid equilibrium dialysis and quantified via LC-MS/MS. Metabolic stability of the compound was determined at a single concentration using species-specific liver microsomes that contained cytochrome P450s, flavin-monooxygenases, carboxyl-esterases, epoxide hydrolases, and other drug-metabolizing enzymes. The quantification was carried out via LC-MS/MS. Membrane permeability of compound 1o was measured using an *in vitro* model for the passive transport from the gastrointestinal tract into the blood system. A double-sink parallel artificial membrane permeability assay PAMPA, in a 96-well format was employed.

**Data availability.** All relevant data are available from the authors on request.

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

## Acknowledgements

The work was supported by the Deutsche Forschungsgemeinschaft (BE 1540/23-1 within SPP 1710 to K.B.), by the Laboratoire d'Excellence (LabEx) ParaFrap (grant LabEx ParaFrap ANR-11-LABX-0024 to E.D.-C. and S.A.B.), by the Equipement d'Excellence (EquipEx) I2MC (grant ANR-11-EQPX-0022 to S.A.B.), by the ERC Starting Grant No 260918 to S.A.B., and by funding from CNRS (E.D.-C. and S.A.B), Inserm (S.A.B), and the University of Strasbourg (E.D.-C. and S.A.B). M.E. and E.D.-C. are grateful to the New York University Abu Dhabi Undergraduate Research Program for a research summer funding to S.N. We also acknowledge Dr Layton Smith and the pharmacology core at Sanford Burnham Prebys Medical Discovery Institute for obtaining the in vitro ADME data.

## Author contributions

R.K. and B.S. synthesized, purified, and characterized compounds of series **1-20**. E.J. and S.R. contributed to the characterization of the effects of the compounds on *Plasmodium* blood stage parasites and to writing the manuscript and preparing the figures. S.W. contributed the experiments in the *P. berghei* mouse model and the speed assay. A.-A.G and S.A.B. characterized the activity of compound **1o** on *P. berghei* ex vivo, on *P. falciparum* gametocytes, and on parasite transmission to mosquitoes. E.D.-C., M.E. and S.N. studied haem interaction and the effects of metals on the activation of the arylmethylamino steroids and analysed the physiochemical data with respect to anti-malarial activities. M.R. characterized the influence of compound **1o** on cellular redox potential. K.V. and A.B. contributed the tests on the antimicrobial profile of the compounds. H.-M.D. determined cytotoxicity and antiproliferative activity on mammalian cells. S.B., T.Q. and C.G.G. characterized the effects of the arylmethylamino steroids on *S. mansoni*. A.B.P. contributed the data on solubility and plasma/microsomal stability in mice and humans. J.B. contributed valuable discussions and comments on the manuscript. K.B. designed and coordinated the study, supervised the experiments on blood-stage parasites, contributed to SAR analyses, and wrote most parts of the manu-script. All authors contributed to figure production and commented on the manuscript.

## Additional information

**Competing financial interests:** The authors declare no competing financial interests.

**Reprints and permission** information is available online at http://npg.nature.com/ reprintsandpermissions/

