## [Peer Review File · Nature Communications]

Reviewers' comments:

Reviewer #1 (Remarks to the Author):

Summary of key results:

The authors report the preparation of a fairly extensive series of about 60 aminosteroids derivatized with various aryl side chains at positions 2, 3, 16 and 17, most with both alpha and beta configurations. The aryl side chains mostly contain hydroxyl or methoxy groups at various positions on the ring. They then report on the in vitro and in vivo antimalarial activities, mammalian cell proliferation inhibition, antibacterial and antischistosomal activities of selected compounds. A number of these compounds show potent selective in vitro antimalarial activity in the low nanomolar range as well as evidence of both i.p. and oral activity in a mouse malaria model. There is clear evidence presented of the ability of some of the derivatives to severely impair *S. mansoni* at 5 μ M concentration. Some preliminary pharmacokinetic data are presented for the most active compound 1o and these indicate relative poor solubility, but good plasma stability and moderate microsomal stability. Finally, the authors investigate the interactions of these compounds with free metal ions, Cu(II) and Fe(III), and with haem. 1o and 2o which have a suitable structure for N,O metal chelation are shown to form fairly stable complexes with Cu(II) and Fe(III). When the OH group is blocked from acting as an O- donor atom by forming a methoxy analogue (1h), such binding is abolished. A series of these compounds were found to be able to complex to haem regardless of whether the O donor was blocked suggesting n-n interactions. However, in the case of compounds 1o and 2o, mass spectral evidence of a covalent adduct with haem was also detected. The authors suggest formation of a quinone methide which reacts with haem to form the adduct.

Originality and Interest:

The finding of potent antimalarial activity in arylaminosteroids is entirely new and this represents a genuinely novel compound class. Indeed, as far as I am aware, very little work has been done on synthetic steroids as antimalarials. A series of steroid tetraoxanes was reported by Milhous and co-workers in 2002 (Solaja et al J. Med. Chem. 45, 3331) and a steroid modified 4-aminoquinoline has also been reported previously (Opsenica et al J. Med. Chem. 2008, 51, 6216). The observation of activity against *Schistosoma* also adds considerable novelty and interest to the work, as does the proposed unique mode of action. These points are definite strengths of this work and would attract significant interest from medicinal chemists in the malaria and schistosomiasis fields, but the work would likely appeal more widely to medicinal chemists, parasitologists and bioinorganic chemists among others.

On the other hand, there are some weaknesses. I must point out, however, that these are not from a technical point of view, since the findings are entirely acceptable as a piece of work in medicinal chemistry. Rather, this relates to the immediate impact and importance of the work as it stands. Firstly, the animal data, while certainly promising, are not outstanding. None of the doses are curative by oral administration. Other compounds have

shown much more promising in vivo data early in the study, for example DDD107498 showed a 90% reduction in parasitaemia at 0.57 mg/kg and was curative as well as exhibiting activity at all parasite stages (Baragana et al Nature, 2015,522,315). So the current compounds, with significant in vivo activity and anti-gametocidal activity and transmission blocking ability don't reach that high bar. The authors acknowledge that further development is needed to bring them to a pre-clinical stage, so it is really a matter of opinion where to set the level of outstanding interest. Secondly, the metal binding and reactivity observations are certainly convincing in the cell-free investigations, but still somewhat speculative as far as the parasite is concerned. Personally, I believe that at least a little direct evidence of the formation of the haem-adduct in the parasite would make the work of sufficient interest for publication.

Data & Methodology:

The work is technically outstanding and I could not find any significant flaws in the analysis, presentation or quality of the work. The authors have been very thorough and the work is well done.

Statistics:

Statistical analysis is of relatively little importance in this work. Nonetheless, more information should be supplied. For example, there are no errors indicated in Table 1. In Figure 2, errors are reported in A, but it is not stated whether these represent SD or SEM and the number of replicates are not stated. There are no error bars shown in B. Error bars are shown and statistical analysis is given in Figure 3, but merits some comment in the figure caption.

Conclusions:

For the most part these are very well supported. The only area of criticism that I have is the somewhat speculative conclusions regarding the metal and haem interactions in the parasite. The logK values for Cu(II) and Fe(III) around 4 indicate that low nM concentrations of compound would be insufficient for the formation of a complex unless the compounds accumulate quite substantially in the parasite. This would need to be shown. Certainly, interest would be greatly enhanced if the authors could provide evidence of the adduct in extracts from treated parasites. This ought to be feasible with mass spectrometric techniques.

References:

The appropriate literature has been included and the work is well referenced.

Clarity and Context:

The work is well written, clear, lucid and easy to follow.

Reviewer #2 (Remarks to the Author):

What are the major claims of the paper?

The main conclusions of the paper are that the group has discovered a novel antimalarial chemotype - steroidal amino cresols; that the compounds are active in vitro and in vivo against both blood stage and sexual stage parasites; and that the mechanism of action involves metal chelation

Are they novel and will they be of interest to others in the field? If the conclusions are not original, it would be helpful if you could provide relevant references.

With respect to the novelty of the compounds, they represent a subclass of aminocresols, which have been well known as antimalarials for decades (examples include WR194965, MK4815, amodiaquine) While it is true that the steroidal aminocresols are novel, it is not clear from the data presented that the steroid contributes significantly to the structure-activity relationships in the series, relative to the cresol. It is clear that the stereochemistry on the D ring matters, but no simplified analogs or detailed structure activity data are presented. It could simply be a complex hydrophobic group.

Is the work convincing, and if not, what further evidence would be required to strengthen the conclusions?

The data supporting the in vitro and in vivo activity of the compounds, and that they are active against both blood and liver stage parasites are convincing. The potency and efficacy are sufficient to place the activity level solidly at that which would be considered for further development. There is some interest in such inhibitors, but they will undoubtedly be conflated with other aminocresols The data supporting the supposition that the compounds act through metal chelation is much more circumstantial and less convincing. Further data supporting formation of such complexes within the parasite and observations of functional consequences of such chemistry are key for this aspect of the work to be convincing. Other approaches might include rescue, quantitative structure activity relationships, etc. Overall this portion of the manuscript is fairly weak.

On a more subjective note, do you feel that the paper will influence thinking in the field?

This manuscript is not likely to significantly move the field as it currently stands. If a novel chelation mechanism could be proven, or at least more strongly supported, then it is more likely it could serve to change people's thinking on amino cresols. None-the-less, there is long term distaste for compounds in that class due to well known issues with toxicology that would lessen enthusiasm.

Reviewer #3 (Remarks to the Author):

Although considerable progress has been made in filling up the antimalarial drug development pipeline, more candidates are always needed and are highly valuable. The characterization of arylmethylamino steroids as highly potent antimalarial agents with in vivo activity and apparent safety is therefore welcome. I can only comment on the biology presented in this manuscript, and offer the following points for the authors to consider.

1. The authors note that the two most potent derivatives against a chloroquine-sensitive strain of *Plasmodium falciparum* were also active against a chloroquine-resistant Dd2 (note: termed K1 in the Materials and Methods section). However, the limited data provided suggest that these compounds may be significantly more potent against the CQ-resistant than the CQ-sensitive strain. This is not unprecedented, having been reported for compounds as diverse as 8-aminoquinolines and amadantine. This deserves further attention (and clarification of the Dd2 vs. K1 discrepancy).
2. Activity in the *P. berghei* model indicates that the compounds have significant, if not single dose cures, efficacy. This model has largely been superseded by a humanized mouse - *P. falciparum* model, which is not always available. Nonetheless, the authors should expound on the differences in the two models and it would be ideal to have the better compounds tested in vivo against *P. falciparum*.
3. Can the authors provide any insight into PK parameters for the active compounds? It is important to be able to correlate in vitro and in vivo data to define exposure-efficacy profiles. Given the apparent high potency and rapid action of these agents in vitro, one might surmise that they are rather quickly cleared from mice, requiring prolonged or high dosing to achieve efficacy. The paper would be much stronger if some indication of in vivo exposure was included.
4. Data for all assays should be reported in consistent units (either molarity or mass/volume).
5. It is not clear that the activity against schistosomes occurs at concentrations below those associated with activity against some microbes and mammalian cells in culture. Is there a therapeutic index? It seems to me that the ~1000-fold less potent schistosomicidal vs. antimalarial activity is telling us that the former activity is likely to be much less interesting or relevant. The authors need to provide convincing evidence that this indication is realistic.
6. I may be showing my ignorance, but note that the authors suggest a quinone-based mechanism of action of these agents. Is there concern about safety in G6PD- patients (as is the case for 8-aminoquinolines)?

Reviewer #1

Summary of key results:

The authors report the preparation of a fairly extensive series of about 60 aminosteroids derivatized with various aryl side chains at positions 2, 3, 16 and 17, most with both alpha and beta configurations. The aryl side chains mostly contain hydroxyl or methoxy groups at various positions on the ring. They then report on the in vitro and in vivo antimalarial activities, mammalian cell proliferation inhibition, antibacterial and antischistosomal activities of selected compounds. A number of these compounds show potent selective in vitro antimalarial activity in the low nanomolar range as well as evidence of both i.p. and oral activity in a mouse malaria model. There is clear evidence presented of the ability of some of the derivatives to severely impair *S. mansoni* at 5 μM concentration. Some preliminary pharmacokinetic data are presented for the most active compound 1o and these indicate relative poor solubility, but good plasma stability and moderate microsomal stability. Finally, the authors investigate the interactions of these compounds with free metal ions, Cu(II) and Fe(III), and with haem. 1o and 2o which have a suitable structure for N,O metal chelation are shown to form fairly stable complexes with Cu(II) and Fe(III). When the OH group is blocked from acting as an O- donor atom by forming a methoxy analogue (1h), such binding is abolished. A series of these compounds were found to be able to complex to haem regardless of whether the O donor was blocked suggesting π - π interactions. However, in the case of compounds 1o and 2o, mass spectral evidence of a covalent adduct with haem was also detected. The authors suggest formation of a quinone methide which reacts with haem to form the adduct.

Originality and Interest:

The finding of potent antimalarial activity in arylaminosteroids is entirely new and this represents a genuinely novel compound class. Indeed, as far as I am aware, very little work has been done on synthetic steroids as antimalarials. A series of steroid tetraoxanes was reported by Milhous and co-workers in 2002 (Solaja et al J. Med. Chem. 45, 3331) and a steroid modified 4-aminoquinoline has also been reported previously (Opsenica et al J. Med. Chem. 2008, 51, 6216). The observation of activity against *Schistosoma* also adds considerable novelty and interest to the work, as does the proposed unique mode of action. These points are definite strengths of this work and would attract significant interest from medicinal chemists in the malaria and schistosomiasis fields, but the work would likely appeal more widely to medicinal chemists, parasitologists and bioinorganic chemists among others.

On the other hand, there are some weaknesses. I must point out, however, that these are not from a technical point of view, since the findings are entirely acceptable as a piece of work in medicinal chemistry. Rather, this relates to the immediate impact and importance of the work as it stands.

Firstly, the animal data, while certainly promising, are not outstanding. None of the doses are curative by oral administration. Other compounds have shown much more promising in vivo data early in the study, for example DDD107498 showed a 90% reduction in parasitaemia at 0.57 mg/kg and was curative as well as exhibiting activity at all parasite stages (Baragana et al Nature, 2015,522,315). So the current compounds, with significant in vivo activity and anti-gametocidal activity and transmission blocking ability don't reach that high bar. The authors acknowledge that

further development is needed to bring them to a pre-clinical stage, so it is really a matter of opinion where to set the level of outstanding interest.

We thank the reviewer for this comment and agree that the “level of outstanding interest” needs to be discussed thoroughly. The aminocresol class has been plagued by a plethora of theoretical safety issues, not least the formation of a putative quinone methide. Interestingly, the actual evidence and data to support such fears are limited. Pyronaridine and amodiaquine are both antimalarial drugs approved by stringent regulatory authorities, which have undergone extensive safety studies in patients without significant findings. Indeed, repeat studies with pyronaridine even resulted in the EMA supporting an improved label based on the well-tolerated findings. Both of these drugs are aminocresols possessing the key theoretical structural liability. Furthermore, as reported in the manuscript, WR194965 was actually studied in humans and apparently was down-prioritised because of the excellent results of mefloquine (with which it was competing) rather than due to safety issues with WR194965. WR194965, obviously, is an aminocresol too. Finally, David Jacobus has several next generation aminocresols that have excellent pharmacokinetics and potency and which are progressing well through preclinical studies. Clearly there have been no safety issues regarding their frontrunners to date. Consequently, the series presented in our manuscript is of interest due to the novelty of the chemistry (necessity of steroid motif), the knowledge that the steroid motif is non-toxic, the rapid action, the effects on gametocytes and transmission (not expected based simply on haemazoin formation), and asexual *in vivo* efficacy. However, as the reviewer pointed out and as stated in the manuscript, the series would need to be optimized based on developability (pharmacokinetics, physico-chemical properties) and studies would need to confirm preclinical safety before a preclinical candidate could be forthcoming. This, however, is in our opinion beyond the scope of this manuscript.

Secondly, the metal binding and reactivity observations are certainly convincing in the cell-free investigations, but still somewhat speculative as far as the parasite is concerned. Personally, I believe that at least a little direct evidence of the formation of the haem-adduct in the parasite would make the work of sufficient interest for publication.

We thank the reviewer for this valuable comment. Unfortunately, detection of haem adducts within the parasite is an intricate and challenging task (particularly within a few months), since the RBCs are rich in hemoglobin catabolites and haem hampers reliable detection and characterization. However, to address the reviewer's comment and further assess haem-adduct formation, we have chosen to confirm their occurrence in cell-free systems by using ESI-MS techniques. We admit that non-covalent complexes cannot be readily differentiated from covalent adducts based solely on *m/z* value or specific fragmentation patterns. However, non-covalent complexes are usually weakly bound and often disassemble into individual components (i.e. drug + haem) upon collision-induced dissociation (CID) in the gas phase. That was clearly confirmed by our first set of MS experiments in which we demonstrated via CID ESI-MS that appreciable amounts of haemadducts (plateau of ~20%) were still observed either with **1o** ($DV_{50} = 290$ V) or **2o** ($DV_{50} = 280$ V) up to 400 V. This first finding suggests stable adduct formation especially when compared to heme itself ($DV_{50} = 329$ V, plateau of ~28 %). It is noteworthy that **1a** and **2a** derivatives that are lacking the phenol amine binding residue lead to haem-adducts with very low abundances and weaker DV_{50} . To go further, methylation of the phenolic position in **1o** and **2o** (i.e. leading to **1h** and **2h**) significantly alters their mass spectrometric properties, and almost no peak or a very weak peaks seemingly arising from haem-adducts were observed significantly. This suggests that the phenol function is of crucial importance in the proposed mechanism and reflects their corresponding antiparasitic activities (see ESI).

To further evidence the covalent character of the haem-adducts observed with **1o** (pseudomolecular mass $m/z = 1,057.7$) or **2o** (pseudomolecular mass $m/z = 1,057.7$), we have considered another set of experiments in which the solutions containing **1o** (or **2o**) and

haem at equimolar concentrations (50 μM) were acidified at very low pH (pH \ll 1) with an excess of TFA. Under these conditions and assuming non-covalent interactions, no haem-adduct formation should be seen due to the unfavorable conditions. Figure SI3c first confirms the data already obtained for the 1:1 stoichiometric mixture of **1o** and haem at pH 7-8, confirming the formation of a haem-adduct at m/z 1,057.7 associated with a high DV_{50} . Importantly, lowering the pH to a value below 1 with an excess of TFA (expressed as pure TFA/mL of solution) does not lead to fading of the peak associated with haem-**1o** species, but on the contrary favors its formation, thus substantiating its covalent nature. The same conclusions can be drawn for compound **2o** with haem (Figure SI3d).

Figure SI3c. ESI mass spectra of a 1:1 mixture of 50 μM haem and 50 μM **1o** in (A) H_2O / CH_3CN (50/50) + 0.1% formic acid and (B) H_2O / CH_3CN (50/50) + 50 $\mu\text{L/mL}$ pure TFA. * = Haem fragmentation products.

Figure SI3d. ESI mass spectra of a 1:1 mixture of 50 μM haem and 50 μM **2o** in (A) H_2O / CH_3CN (50/50) + 0.1% formic acid and (B) H_2O / CH_3CN (50/50) + 50 $\mu\text{L/mL}$ pure TFA. * = Haem fragmentation products.

To go further, we performed the same experiments on a steroid derivative (designated as **STS948**, see Figure 1 in the manuscript) that displays a steroid moiety together with nitrogen-based units that are known to bind haem. In contrast to **1o** and **2o**, this compound is clearly not prone to quinone methide formation and according to our hypothesis would not lead to covalent drug-haem adducts. At pH 7.5 and using absorption spectrophotometric titrations, **STS948** was shown to bind haem with a K_D (0.8 μM) larger than those measured with **1o** (3.3 μM) or **2o** (8.3 μM) (Figure SI3e). CID ESI-MS conducted at pH 7-8 is further

evidence of the presence of a **STS948**:haem adduct associated to a DV_{50} of 367 V along with 36% of complex remaining at a fragmentor voltage of 400 V (Figure SI3f). At a first sight, these data would suggest a more active antiplasmodial derivative based on the data already obtained for **1o** and **2o** and therefore invalidate our initial hypotheses. However, the antiplasmodial activity IC_{50} on the *Plasmodium falciparum* 3D7 strain was found to be 675 nM, far behind the nM antiplasmodial activities measured for **1o** and **2o**.

Figure SI3e. (A) Spectrophotometric titration of haem (under its π - π dimeric form, $\log K_{Dim} = 6.82$) by **STS948**, and (B) electronic spectra of haem, the free substrates, and its corresponding Fe^{III} PPIX substrate complex with **STS948**. Solvent: 0.2 M aqueous sodium HEPES buffer pH 7.5; $T = 25.0$ °C; $l = 1$ cm. For **STS948**: $[Fe^{III}PPIX]_0 = 2.41 \times 10^{-5}$ M; (1) $[STS948]_0/[Fe^{III}PPIX]_0 = 0$; (2) $[STS948]_0/[Fe^{III}PPIX]_0 = 2.78$. $\log K_{Fe^{III}PPIX,STS948} = 6.09 \pm 0.06$.

Figure SI3f. (A) ESI mass spectrum of a 1:1 mixture of 50 μ M haem and 50 μ M **STS948** in H_2O/CH_3CN (50/50) + 0.1% formic acid. (B) Stability responses of the haem substrate adduct obtained from ESI-MS-CID experiments. Positive mode; fragmentor voltage from 20 V to 400 V with 20 V increments. * = $[(haem)_2-H]^+$ ($m/z = 1,231.65$); ** = $[(haem)_2+OH]^+$ ($m/z = 1,249.65$).

To get a deeper insight into this haem-binding process, we performed the same set of MS experiments under classic conditions (CH_3CN/H_2O + 0.1% formic acid pH 7-8, Figure SI3g) and under acidic conditions. If the first experiments carried out at neutral pH corroborate the formation of **STS948**:haem adduct at m/z 1,128.75, those conducted under acidic conditions reveal the absence of any complexation between **STS948** and haem. These data clearly

confirmed the expected non-covalent character charring between **STS948** and haem and give credibility to our assumptions on covalent complexes with **1o** and **2o**.

Figure S13g. ESI mass spectra of a 1:1 mixture of 50 μM haem and 50 μM **STS948** in (A) $\text{H}_2\text{O}/\text{CH}_3\text{CN}$ (50/50) + 0.1% formic acid and (B) $\text{H}_2\text{O}/\text{CH}_3\text{CN}$ (50/50) + 50 $\mu\text{L}/\text{mL}$ pure TFA. * = Haem fragmentation products.

Data & Methodology:

The work is technically outstanding and I could not find any significant flaws in the analysis, presentation or quality of the work. The authors have been very thorough and the work is well done.

We thank the reviewer for this comment!

Statistics:

Statistical analysis is of relatively little importance in this work. Nonetheless, more information should be supplied. For example, there are no errors indicated in Table 1.

Some of the mean values reported in Table 1 (e.g. cytotoxicity) are based on three single values only, which does not really allow for statistical analysis. Other values in the manuscript/table, e.g. *Plasmodium* cell culture data, have been repeated several times with several technical replicates each. We therefore would prefer not to indicate errors in the table and rather state: "All values are mean values of at least three independent determinations that differed by less than 20%."

In Figure 2, errors are reported in A, but it is not stated whether these represent SD or SEM and the number of replicates are not stated.

Three biological replicates were performed and SDs were used. This has been added to the manuscript.

There are no error bars shown in Figure 2B.

Error bars have been added.

Error bars are shown and statistical analysis is given in Figure 3, but merits some comment in the figure caption.

We have updated the legend of Figure 3 to include a description of the error bars. The statistical analyses performed for the experiments reported in Figure 3 are briefly mentioned in the legend and are described in detail in the Material and Methods section. We have added one sentence at the end of this section to give some additional precision on the statistical models we used.

Conclusions:

For the most part these are very well supported. The only area of criticism that I have is the somewhat speculative conclusions regarding the metal and haem interactions in the parasite. The logK values for Cu(II) and Fe(III) around 4 indicate that low nM concentrations of compound would be insufficient for the formation of a complex unless the compounds accumulate quite substantially in the parasite. This would need to be shown. Certainly, interest would be greatly enhanced if the authors could provide evidence of the adduct in extracts from treated parasites. This ought to be feasible with mass spectrometric techniques.

We agree with this reviewer on the fact that a log K of about 4 is a rather low stability constant. However, one should keep in mind the scientific context as well as the fact that red blood cells (RBCs) represent a hemoglobin-rich (and iron-rich) environment. It is usually assumed that for healthy RBCs, the hemoglobin mass concentrations are at about 12-18 grams dL^{-1} , which corresponds to hemoglobin concentrations of about 10 mM. It is also accepted that the erythrocytic forms of *Plasmodium* degrade about 60% to 80% of the total RBC Hb content as a vital source of amino acids (see Francis SE, Sullivan DJ Jr, Goldberg DE (1997). Hemoglobin metabolism in the malaria parasite *Plasmodium falciparum*. *Ann. Rev. Microbiol.* 51:97–123). As a consequence, parasitized RBCs contain high amounts of haem and free iron. We demonstrated in the manuscript that compound **1o** displayed good stability in plasma (62% and 58% remaining, respectively, after a 3-hour incubation with human and mouse plasma) with however a poor aqueous solubility ($< 1 \mu\text{g/ml}$ across a range of pH; $< 2.3 \times 10^{-6} \text{ M}$). Assuming a concentration of 10^{-8} M of **1o** (100 times lower than the aqueous solubility) and $7 \times 10^{-4} \text{ M}$ (10% of the degraded Hb under the form of free Fe), one can estimate that more than 80% of **1o**:Fe complex will be formed. In the case of haem complexes with **1o** ($K_D \sim \mu\text{M}$), much higher amounts of the haem:**1o** adduct will be formed. This rather simple approach demonstrates that despite the low stability of the metal complexes with **1o**, a high percentage of the complexes are expected to be formed due to the peculiar conditions of *Plasmodium falciparum*-infected RBCs. Last but not least, due to the aforementioned characteristics of *Plasmodium falciparum*-infected RBCs, it is very difficult to analyze complexes with **1o** from Fe- and haem-enriched environments within the time provided for revision.

These considerations as well as Figure SI3i have been added to the manuscript.

Figure SI3i. Species diagram distribution showing the formation of **1o.Fe** complexes within parasitized RBCs and assuming relevant Hb catabolites and **1o** parameters.

References:

The appropriate literature has been included and the work is well referenced.

Clarity and Context:

The work is well written, clear, lucid and easy to follow.

Reviewer #2 (Remarks to the Author):

What are the major claims of the paper?

The main conclusions of the paper are that the group has discovered a novel antimalarial chemotype - steroidal amino cresols; that the compounds are active in vitro and in vivo against both blood stage and sexual stage parasites; and that the mechanism of action involves metal chelation.

Are they novel and will they be of interest to others in the field? If the conclusions are not original, it would be helpful if you could provide relevant references.

With respect to the novelty of the compounds, they represent a subclass of aminocresols, which have been well known as antimalarials for decades (examples include WR194965, MK4815, amodiaquine) While it is true that the steroidal aminocresols are novel, it is not clear from the data presented that the steroid contributes significantly to the structure-activity relationships in the series, relative to the cresol. It is clear that the stereochemistry on the D ring matters, but no simplified analogs or detailed structure activity data are presented. It could simply be a complex hydrophobic group.

We thank the reviewer for this comment and agree that the contribution of the steroid moiety to the biological activity is an important aspect. We therefore have revised the manuscript as follows:

Figure 1a has been changed. The structures of the non-steroid compounds tested are now also provided, and their synthesis has been incorporated into the supplementary information.

The IC₅₀ values of the non-steroid compounds have been added to Table SI2a.

The text has been revised as follows: “*De facto*, none of the non-steroid-derived analogs (series **15-20**, Table SI2a) exhibited antimalarial activity comparable to the steroid compounds. The IC₅₀ values on *P. falciparum* 3D7 asexual blood stages were 2 μM for **17c**, 4 μM for **18c**, 5.6 μM for **20c**, >6 μM for **15c** and **16c**, and >15 μM for **19c**. Notably, compound **18c** is derived from the lipophilic 1-adamantylamine which is closely related to the neuroprotective agent *memantine* (3,5-dimethyl-1-adamantylamine) (Stuart, J. T. & Grossberg, G. T. Memantine: a review of studies into its safety and efficacy in treating Alzheimer’s disease and other dementias. *Clin. Interv. Aging.* **4**, 367–377 (2009)). Based on these data, the steroid part is essential for the biological activity of the compounds. This interesting new chemical entity seems to vectorize the aminocresols to parasitized red blood cells and is potentially recognized by transporters. The present study might therefore initiate new investigations based on the use of steroids to target not only drugs but also fluorophores or other chemical probes to cells of interest.”

Is the work convincing, and if not, what further evidence would be required to strengthen the conclusions?

The data supporting the in vitro and in vivo activity of the compounds, and that they are active against both blood and liver stage parasites are convincing. The potency and efficacy are sufficient to place the activity level solidly at that which would be considered for further development. There is some interest in such inhibitors, but they will undoubtedly be conflated with other aminocresols. The data supporting the supposition that the compounds act through metal chelation is much more circumstantial and less convincing. Further data supporting formation of such complexes within the parasite and observations of functional consequences of such chemistry are key for this aspect of the work to be convincing. Other approaches might include rescue, quantitative structure activity relationships, etc. Overall this portion of the manuscript is fairly weak.

As indicated above (please see answer to comment 1 of reviewer 1), additional experiments have been performed to further prove the presence of metal and haem complexes. Furthermore, theoretical considerations concerning the *in vivo* situation have been added to the manuscript (please see answer to conclusions of reviewer 1). All new data and calculations have been incorporated into the manuscript.

The former Figure 5 showing the potential mechanism of action has been omitted.

The former Figure 6 (now Figure 5) has been revised.

The following paragraph has been added: “All these findings make a quinone methide mechanism permitted by R very likely. Furthermore, *ortho* phenols are most efficient while their methyl ethers as well as *meta*-substituted phenols are less active (Figure 5). This fact supports the participation of a chelation mechanism, including activation processes and ongoing redox processes mediated by metals. The recovery of antimalarial activity from *meta*- to *para*-substituted phenols suggests that additional mechanisms are likely to support activity, as the latter cannot act as bidentate ligands.

On a more subjective note, do you feel that the paper will influence thinking in the field?

This manuscript is not likely to significantly move the field as it currently stands. If a novel chelation mechanism could be proven, or at least more strongly supported, then it is more likely it could serve to change people's thinking on amino cresols. None-the-less, there is long term distaste for compounds in that class due to well known issues with toxicology that would lessen enthusiasm.

Concerning the novelty of the compounds and the fact that they represent a subclass of aminocresols, please see our answer to comment 1 of reviewer 1:

“...We thank the reviewer for this comment and agree that the “level of outstanding interest” needs to be discussed thoroughly. The aminocresol class has been plagued by a plethora of theoretical safety issues, not least the formation of a putative quinone methide. Interestingly, the actual evidence and data to support such fears are limited. Pyronaridine and amodiaquine are both antimalarial drugs approved by stringent regulatory authorities, which have undergone extensive safety studies in patients without significant findings. Indeed, repeat studies with pyronaridine even resulted in the EMA supporting an improved label based on the well-tolerated findings. Both of these drugs are aminocresols possessing the key theoretical structural liability. Furthermore, as reported in the manuscript, WR194965 was actually studied in humans and apparently was down-prioritised because of the excellent results of mefloquine (with which it was competing) rather than due to safety issues with WR194965. WR194965, obviously, is an aminocresol too. Finally, David Jacobus has several next generation aminocresols that have excellent pharmacokinetics and potency and which are progressing well through preclinical studies. Clearly there have been no safety issues regarding their frontrunners to date. Consequently, the series presented in our manuscript is of interest due to the novelty of the chemistry (necessity of steroid motif), the knowledge that the steroid motif is non-toxic, the rapid action, the effects on gametocytes and transmission (not expected based simply on haemazoin formation), and asexual *in vivo* efficacy. However, as the reviewer pointed out and as stated in the manuscript, the series would need to be optimized based on developability (pharmacokinetics, physico-properties) and studies would need to confirm preclinical safety before a preclinical candidate could be forthcoming. This, however, is in our opinion beyond the scope of this manuscript...”

It should further be noted that there has been a renaissance in the use of specific covalent binding agents as small molecule therapeutics despite inherent prejudices based on structures. Provided that covalent binding only occurs at the site of pharmacological interest, or that formation of a covalent adduct (that can then bind to the target of interest) is only formed in the correct organelle, then safety issues associated with covalency are significantly reduced. This is exactly as has been shown in regard to covalent kinase inhibitors in cancer (e.g. EGFR inhibitors). We postulate that formation of a quinone methide could lead to a covalent interaction with the parasite. Given the measured effects on haemazoin formation, the lipophilicity of the molecules, and their basic nature, it is most likely that they access the food vacuole of the parasite. At the pH of the organelle (5.0) these molecules will be protonated and concentrated. Protonation of an aminocresol will increase the potential for a site-specific formation of a reactive species, limiting any safety risk. Furthermore, the SAR demonstrates that compounds not having an aminocresol retain potency (albeit reduced), suggesting that the structures retain other pharmacology unrelated to this covalent binding; this is an innovation – delivering a compound having polypharmacology – with the benefits such as resistance selection apparent.”

Parts of these considerations have been incorporated into the manuscript.

Reviewer #3 (Remarks to the Author):

Although considerable progress has been made in filling up the antimalarial drug development pipeline, more candidates are always needed and are highly valuable. The characterization of arylmethylamino steroids as highly potent antimalarial agents with *in vivo* activity and apparent safety is therefore welcome. I can only comment on the biology presented in this manuscript, and offer the following points for the authors to consider.

1. The authors note that the two most potent derivatives against a chloroquine-sensitive strain of *Plasmodium falciparum* were also active against a chloroquine-resistant Dd2 (note: termed K1 in the Materials and Methods section). However, the limited data provided suggest that these compounds may be significantly more potent against the CQ-resistant than the CQ-sensitive strain. This is not unprecedented, having been reported for compounds as diverse as 8-aminoquinolines and amantadine. This deserves further attention (and clarification of the Dd2 vs. K1 discrepancy).

We thank the reviewer for this comment. In fact, the CQ-resistant Dd2 strain was used for these analyses. The typing error has been corrected. A figure (new Figure SI2a) showing representative IC₅₀ curves for compound **2o** is now provided, and the discussion of the data has been expanded.

2. Activity in the *P. berghei* model indicates that the compounds have significant, if not single dose cures, efficacy. This model has largely been superseded by a humanized mouse - *P. falciparum* model, which is not always available. Nonetheless, the authors should expound on the differences in the two models and it would be ideal to have the better compounds tested *in vivo* against *P. falciparum*.

We thank the reviewer for this comment! The Swiss Tropical and Public Health Institute has the humanized mouse model in place. However, presently it is exclusively used for MMV projects. Furthermore it is quite time consuming and costly. It will definitely be of great interest to test the best steroid compounds – or their next generation – in the humanized mouse model in the future. However, we believe that these studies are beyond the scope of the present manuscript, which contains data from chemistry to *in vivo* studies using the *P. berghei* model.

3. Can the authors provide any insight into PK parameters for the active compounds? It is important to be able to correlate *in vitro* and *in vivo* data to define exposure-efficacy profiles. Given the apparent high potency and rapid action of these agents *in vitro*, one might surmise that they are rather quickly cleared from mice, requiring prolonged or high dosing to achieve efficacy. The paper would be much stronger if some indication of *in vivo* exposure was included.

In response to reviewer 3, we have determined the mouse PK parameters for compound **1o** and included them in the manuscript. The compound has good exposure after *i.p.* dosing but relatively low exposure after oral dosing, which helps put the *in vivo* efficacy results in context. The data have been included as Table SI2e in the supplementary material section. This has been included in the text: “Moreover, we performed pharmacokinetics studies in mice, dosing *i.v.*, *i.p.*, and *p.o.* (Table SI2e). **1o** displayed moderate clearance (29.6 ml/min/kg) with a long half-life (> 8 hr) after *i.v.* dosing. The compound showed modest oral bioavailability (%F < 5) but sustained plasma levels of ~100 nM after oral dosing at

100 mg/kg. In contrast, high exposure and sustained levels of ~2 μM were obtained after i.p. dosing at 100 mg/kg.”

4. Data for all assays should be reported in consistent units (either molarity or mass/volume).

We thank the reviewer for this comment! Data for all assays are now reported in molar concentrations. Tables, figures, and text have been changed accordingly.

5. It is not clear that the activity against schistosomes occurs at concentrations below those associated with activity against some microbes and mammalian cells in culture. Is there a therapeutic index? It seems to me that the ~1000-fold less potent schistosomicidal vs. antimalarial activity is telling us that the former activity is likely to be much less interesting or relevant. The authors need to provide convincing evidence that this indication is realistic.

There is quite a difference between targeting a single cell organism and a multicellular organism. One consequence of this fact is that the complex physiology of a multicellular organism requires the use of a higher concentration of any compound that may exert effects on a single-cell organism at a lower concentration.

We found clear effects on adult schistosomes *in vitro* with arylmethylamino steroids at concentrations of 5-10 μM . This is exactly in the range of the gold standard praziquantel, which is toxic for schistosomes at 5-10 μM but no longer at 1 μM *in vitro* [1, 2, and many older studies]. Furthermore, compounds tested today *in vitro* for their suitability as lead structures for further drug development also showed anti-schistosomal activities in this concentration range of 1.4-9.5 μM (malaria box containing 200 diverse drug-like compounds; [2]).

There is no calculated therapeutic index yet, and it may be too early for speculations in this phase of finding lead compounds for further development. However, based on the cytotoxicity results obtained so far, there are reasonable selectivity indices ≥ 10 , which correspond to those found in similar studies with new compounds (1.18-40.00; [2]). With praziquantel, selectivity indices are also not extremely different and vary with the cell line used for comparison. E.g., for a hepatic cell line (HepG2), 165 μM (IC_{50}) were found to be toxic, which corresponds to a selectivity index of 33 (16.5) with respect to the IC_{50} of adult worms treated *in vitro* with 5 μM (10 μM) [3]. These facts support our conclusion that arylmethylamino steroids are interesting lead candidates and realistic options for further development as anti-schistosomal compounds.

[1] Magalhães LG, Machado CB, Morais ER, Moreira EB, Soares CS, da Silva SH, Da Silva Filho AA, Rodrigues V (2009) *In vitro* schistosomicidal activity of curcumin against *Schistosoma mansoni* adult worms. *Parasitol Res* 104(5):1197-201

[2] Ingram-Sieber K, Cowan N, Panic G, Vargas M, Mansour NR, Bickle QD, Wells TN, Spangenberg T, Keiser J (2014) Orally active antischistosomal early leads identified from the open access malaria box. *PLoS Negl Trop Dis* 8(1):e2610.

[3] Sun Q, Mao RF, Wang DL, Hu CY, Zheng Y, Sun DQ (2016) The cytotoxicity study of praziquantel enantiomers. *Drug Des Devel Ther* 10, 2061-2068

This information has been added to the manuscript in condensed form.

6. I may be showing my ignorance, but note that the authors suggest a quinone-based mechanism of action of these agents. Is there concern about safety in G6PD- patients (as is the case for 8-aminoquinolines)?

Aminocresols are suggested to act through a quinone-based mechanism of action, making

likely the generation of reactive oxygen species (ROS), as in the case of 8-aminoquinolines exemplified by primaquine. As pointed out by the reviewer, drugs acting as ROS inducers bear a hemolytic risk in populations with G6PD deficiency when used for a long term and at high doses. Primaquine is currently underutilized despite its great potential to reduce malaria transmission and contain artemisinin resistance. However, there is a renewal of interest in primaquine because it is the only available drug with established activity against mature gametocytes. The drug is currently under intense clinical validation for widespread use in combination with ACTs, at low dose, and in short-term treatment under conditions of safety for G6PD patients. Primaquine is a pro-oxidative drug, although the molecular mechanism underlying the redox-related killing of gametocytes is poorly understood and hampers the rational design of new gametocytocidal compounds. We believe that pro-oxidant drugs need to be investigated further to find new companions of ACTs besides the use of primaquine.

Another prominent prooxidant antimalarial that is presently intensely discussed is methylene blue. In spite of its strong redox cycling activity, methylene blue – given as a 3-day oral combination regimen of a total of 1,500 mg CQ and 780 mg MB – was shown to be well tolerated by healthy adult men with G6PD deficiency (Mandi et al., Safety of the combination of chloroquine and methylene blue in healthy adult men with G6PD deficiency from rural Burkina Faso. *Trop. Med. Int. Health* 10, 32-38 (2005).

In order to address the question of the reviewer and to further study the potentially redox-related mechanism of action of the steroid compounds, compound **1o** was tested in cell culture on the chloroquine-sensitive *P. falciparum* 3D7 strain expressing the cytosolic glutathione redox sensor hGrx1-roGFP2. As indicated in the newly inserted Figure SI2b, 24 h incubation with 20, 50, and 100 nM compound **1o** led to a dose-dependent increase of the redox ratio of the probe, pointing to increased oxidation and alterations in the intracellular redox potential. Although these results fell short in reaching significance in the experimental setup chosen, they point towards a potential influence of compound **1o** on glutathione redox homeostasis, which deserves to be studied in more detail.

REVIEWERS' COMMENTS:

Reviewer #1 (Remarks to the Author):

The authors have considerably strengthened the manuscript with additional experiments which address most of the concerns that I had with the original manuscript. The addition of non-steroid cresols that nonetheless bear hydrophobic groups provides good evidence for the requirement of the steroid, particularly the comprehensive list of activities available for derivative c, which shows weak activity for all of compounds 15 – 20 and some of the steroid derivatives, but very strong activity for some of the steroid derivatives. This seems to be quite compelling. Given the role of the steroid component, together with the significant activity against *S. mansoni*, the manuscript justifies its novelty.

Regarding the role of metal ions and haem in activity, the authors have addressed the problem in two ways. They have obtained additional evidence of haem adducts of 1o and 2o using mass spectrometry. They have further shown that this adduct formation is not subject to proton competition at low pH, which is compelling evidence that they are not dealing with a coordination complex. Indeed, this is supported by the very nice experiment performed with STS 948. Incidentally, the spectrum of the STS 948 complex with Fe(III)PPIX is a very typical example of a low spin complex such as one would see with either pyridine or imidazole. Given that this compound has side chains with both moieties, it is compelling evidence for coordination of one of these groups to the Fe(III) centre. In this case, as expected protonation of the ligand at low pH displaces it from Fe(III)PPIX as seen in the mass spectra. While this provides good evidence of haem adduct formation in cell free laboratory conditions, it does not necessarily prove that it forms in the parasite. That being said, the second way in which the authors have addressed the issue is by making it clearer that they are proposing haem adduct formation as an hypothesis, rather than a definite mechanism. I agree that detecting small quantities of one haem species against a large background of haemoglobin is difficult and would require an extensive study in its own right. Their arguments in respect of the strength of interaction with haem and degree of complex formation are correct in the sense that haem concentrations are at least nominally high (they may not actually be so high in solution, since haematin has very low solubility at low pH). My main point, however, was not around the haem interaction, but rather with respect to complex formation with Cu(II) and Fe(III) which have much lower intracellular concentrations as free ions (as opposed to Fe(III)PPIX). Possibly the explanation for the increased efficacy of the ortho substituted derivatives is that transient complexes are involved in generation of radicals in a manner somewhat analogous to what is proposed in the case of bleomycin. The one additional experiment that the authors could have undertaken is to see whether chelating agents that form redox inactive chelates decrease the activity of these compounds. That would make the metal ion story much more compelling and would be a relatively easy experiment to perform.

The addition PK experiments go a long way towards explaining the relatively weak in vivo oral activity, but of course do raise another question. Is the poor p.o. activity a result of poor absorption in the intestine, or is it a result of extensive first pass metabolism resulting in removal of the steroid group via an N-dealkylation process? I don't think it is the focus of

the current study, but might be worth a comment for future studies.

Overall, the work is now quite compelling in my view.

Reviewer #2 (Remarks to the Author):

The authors have successfully addressed all of the criticisms raised on earlier review.

The addition of data solidifying the claims about chelation and covalent complex formation are particularly critical and significantly add to the defensible conclusions.

The pharmacokinetics experiments are also very important to allow interpretation of the efficacy experiments; however they highlight another issue -- the molecule does not have sufficient oral bioavailability to be considered a candidate under current criteria.

The arguments concerning toxicology add some to the work, but the potential liability remains. One will not know the significance of this risk until detailed in vivo toxicology studies are carried out. From this reviewer's point of view this is a neutral issue with respect to publication. The authors should exert some caution, however, in fully extrapolating from the examples they choose -- there are others where the liability is significant.

Reviewer #3 (Remarks to the Author):

The authors have answered my concerns thoroughly and convincingly, and the material added in response to concerns of the other reviewers has greatly strengthened the manuscript. I believe that further revision is unnecessary and that the manuscript should be processed for publication.

We would like to thank the editor, the journal staff and all reviewers for their fruitful and constructive comments!

Reviewer #1

The authors have considerably strengthened the manuscript with additional experiments which address most of the concerns that I had with the original manuscript. The addition of non-steroid cresols that nonetheless bear hydrophobic groups provides good evidence for the requirement of the steroid, particularly the comprehensive list of activities available for derivative **c**, which shows weak activity for all of compounds 15 – 20 and some of the steroid derivatives, but very strong activity for some of the steroid derivatives. This seems to be quite compelling. Given the role of the steroid component, together with the significant activity against *S. mansoni*, the manuscript justifies its novelty.

Regarding the role of metal ions and haem in activity, the authors have addressed the problem in two ways. They have obtained additional evidence of haem adducts of **1o** and **2o** using mass spectrometry. They have further shown that this adduct formation is not subject to proton competition at low pH, which is compelling evidence that they are not dealing with a coordination complex. Indeed, this is supported by the very nice experiment performed with STS 948. Incidentally, the spectrum of the STS 948 complex with Fe(III)PPIX is a very typical example of a low spin complex such as one would see with either pyridine or imidazole. Given that this compound has side chains with both moieties, it is compelling evidence for coordination of one of these groups to the Fe(III) centre. In this case, as expected protonation of the ligand at low pH displaces it from Fe(III)PPIX as seen in the mass spectra. While this provides good evidence of haem adduct formation in cell free laboratory conditions, it does not necessarily prove that it forms in the parasite. That being said, the second way in which the authors have addressed the issue is by making it clearer that they are proposing haem adduct formation as an hypothesis, rather than a definite mechanism. I agree that detecting small quantities of one haem species against a large background of haemoglobin is difficult and would require an extensive study in its own right. Their arguments in respect of the strength of interaction with haem and degree of complex formation are correct in the sense that haem concentrations are at least nominally high (they may not actually be so high in solution, since haematin has very low solubility at low pH). My main point, however, was not around the haem interaction, but rather with respect to complex formation with Cu(II) and Fe(III) which have much lower intracellular concentrations as free ions (as opposed to Fe(III)PPIX). Possibly the explanation for the increased efficacy of the ortho substituted derivatives is that transient complexes are involved in generation of radicals in a manner somewhat analogous to what is proposed in the case of bleomycin. The one additional experiment that the authors could have undertaken is to see whether chelating agents that form redox inactive chelates decrease the activity of these compounds. That would make the metal ion story much more compelling and would be a relatively easy experiment to perform.

We thank the reviewer for the excellent comment about the generation of radicals as the most probable explanation for the increased efficacy of the ortho-substituted derivatives (1o and derivatives found in the present work). We added the following paragraph to the manuscript on page 10: „Interestingly, the most potent antimalarial compounds (1o and analogues found in the present work) are the ortho-substituted derivatives, suggesting that generation of radicals is the most probable explanation for the increased efficacy of the chemical series. Similar ortho-substituted ligands of transient metal complexes, such as the reduced benzoylmenadione metabolite of the

antimalarial plasmodione, which presents an oxygen-rich bidentate site suitable for Fe(III) chelation (Bielitza et al, 2015), were reported to become redox-inactive when their effects were antagonized with the known iron(III) chelator desferoxamine (DFO). In this study a most potent antagonistic effect (sum FIC₅₀ = 2.4) induced by DFO was observed when combined with plasmodione in cell culture assays using RBCs parasitized by P. falciparum 3D7.

Bielitza, M., Belorgey, D., Ehrhardt, K., Johann, L., Lanfranchi, D.A., Gallo, V., et al. Antimalarial NADPH-consuming redox-cyclers as superior glucose-6-phosphate dehydrogenase deficiency copycats. Antioxid Redox Signal. , 22, 1337-51 (2015).

This reference has been inserted into the manuscript. All following references have been renumbered.

The addition PK experiments go a long way towards explaining the relatively weak in vivo oral activity, but of course do raise another question. Is the poor p.o. activity a result of poor absorption in the intestine, or is it a result of extensive first pass metabolism resulting in removal of the steroid group via an N-dealkylation process? I don't think it is the focus of the current study, but might be worth a comment for future studies.

We are unsure what is the cause of the poor oral bioavailability. We hypothesize that the absorption could be poor based on the low permeability observed in the PAMPA assay, although as we note the compound was trapped in the filter membrane in the assay. It is most likely not solely first pass metabolism based on the moderate microsomal stability we measured. Also, we have not performed metabolite ID studies as this is beyond the scope of the current studies. While the particular leads described in the paper are not potential clinical candidates, we believe they are valuable tool compounds and could serve as points for further optimization. The fact that oral bioavailability needs to be further improved is now further emphasized in the discussion: „The next steps for further optimizing our steroid compounds include in-depth analyses of pharmacokinetic and toxicological properties, further improvement of lipophilicity and ADME properties including oral bioavailability and enhancement of potency in vivo.“

Overall, the work is now quite compelling in my view.

Reviewer #2

The authors have successfully addressed all of the criticisms raised on earlier review.

The addition of data solidifying the claims about chelation and covalent complex formation are particularly critical and significantly add to the defensible conclusions.

The pharmacokinetics experiments are also very important to allow interpretation of the efficacy experiments; however they highlight another issue -- the molecule does not have sufficient oral bioavailability to be considered a candidate under current criteria.

We are unsure what is the cause of the poor oral bioavailability. We hypothesize that the absorption could be poor based on the low permeability observed in the PAMPA assay, although as we note the compound was trapped in the filter membrane in the

assay. It is most likely not solely first pass metabolism based on the moderate microsomal stability we measured. Also, we have not performed metabolite ID studies as this is beyond the scope of the current studies. While the particular leads described in the paper are not potential clinical candidates, we believe they are valuable tool compounds and could serve as points for further optimization. The fact that oral bioavailability needs to be further improved is now further emphasized in the discussion: „The next steps for further optimizing our steroid compounds include in-depth analyses of pharmacokinetic and toxicological properties, further improvement of lipophilicity and ADME properties including oral bioavailability and enhancement of potency in vivo.“

The arguments concerning toxicology add some to the work, but the potential liability remains. One will not know the significance of this risk until detailed in vivo toxicology studies are carried out. From this reviewer's point of view this is a neutral issue with respect to publication. The authors should exert some caution, however, in fully extrapolating from the examples they choose -- there are others where the liability is significant.

We thank the reviewer for this comment and agree that extrapolations can be done with greatest caution only. The sentence on page 14 has been adapted: „However, more extensive and thorough testing will be necessary during further preclinical development of one of the two best compounds or optimized analogues.“

Reviewer #3

The authors have answered my concerns thoroughly and convincingly, and the material added in response to concerns of the other reviewers has greatly strengthened the manuscript. I believe that further revision is unnecessary and that the manuscript should be processed for publication.

We thank the reviewer for this comment!